# Mistargeting of aggregation prone mitochondrial proteins activates a nucleus-mediated posttranscriptional quality control pathway in trypanosomes

Caroline E. Dewar[1], Silke Oeljeklaus [2,3], Jan Mani[1], Wignand W. D. Mühlhäuser[3], Corinne von Känel[1], Johannes Zimmermann [2], Torsten Ochsenreiter[4], Bettina Warscheid [2,3,5✉] & André Schneider [1✉]

Mitochondrial protein import in the parasitic protozoan *Trypanosoma brucei* is mediated by the atypical outer membrane translocase, ATOM. It consists of seven subunits including ATOM69, the import receptor for hydrophobic proteins. Ablation of ATOM69, but not of any other subunit, triggers a unique quality control pathway resulting in the proteasomal degradation of non-imported mitochondrial proteins. The process requires a protein of unknown function, an E3 ubiquitin ligase and the ubiquitin-like protein (TbUbL1), which all are recruited to the mitochondrion upon ATOM69 depletion. TbUbL1 is a nuclear protein, a fraction of which is released to the cytosol upon triggering of the pathway. Nuclear release is essential as cytosolic TbUbL1 can bind mislocalised mitochondrial proteins and likely transfers them to the proteasome. Mitochondrial quality control has previously been studied in yeast and metazoans. Finding such a pathway in the highly diverged trypanosomes suggests such pathways are an obligate feature of all eukaryotes.

[1] Department of Chemistry, Biochemistry and Pharmaceutical Sciences, University of Bern, Freiestrasse 3, Bern CH-3012, Switzerland. [2] Faculty of Chemistry and Pharmacy, Department of Biochemistry, Theodor Boveri-Institute, University of Würzburg, 97074 Würzburg, Germany. [3] Biochemistry and Functional Proteomics, Institute of Biology II, Faculty of Biology, University of Freiburg, 79104 Freiburg, Germany. [4] Institute of Cell Biology, University of Bern, Baltzerstrasse 4, Bern CH-3012, Switzerland. [5] CIBSS Centre for Integrative Biological Signalling Studies, University of Freiburg, 79104 Freiburg, Germany.
✉email: Bettina.Warscheid@biologie.uni-freiburg.de; andre.schneider@dcb.unibe.ch

Mitochondria originally arose due to a single endosymbiotic event between an alpha-proteobacterium and an archaeal host cell[1,2]. Subsequently, the endosymbiont evolved into a nuclearly-controlled organelle by transferring most of its genes to the host genome, and by acquiring pathways allowing the import of the now cytosolically-synthesised proteins. While the mitochondrion has retained its own genome, it encodes only a small number of genes. As a consequence, more than 95% of mitochondrial proteins are synthesised in the cytosol and need to be imported and sorted to the correct intra-organellar compartments[3–5].

The mitochondrion is essential for the viability of nearly all eukaryotes. Its main function is the generation of ATP by oxidative phosphorylation, which is mediated by large inner membrane-integrated protein complexes consisting of both imported and mitochondrially-synthesised proteins. Mitochondria also contain other important pathways including iron-sulfur cluster biogenesis, which provides essential co-factors for numerous mitochondrial, cytosolic and nuclear proteins[6].

Mitochondrial dysfunction not only harms the organelle but can cause collateral damage to the whole cell by disturbing cellular proteostasis. Recently, it became clear that impairment of mitochondrial protein import is a major factor contributing to this[7–9]. Conditions challenging protein import include mutations in the subunits of the protein translocases, mistargeted or misfolded substrates, and the stalling of the import channel by precursor overload or tightly folded precursors. Moreover, low ATP levels, a reduced mitochondrial membrane potential and disturbance of the coordination between nuclear and organellar gene expression may also influence protein import. It is therefore not surprising that eukaryotes evolved mitochondrial quality control (MQC) pathways that prevent or deal with organellar dysfunction and its consequences for the cell[8–12]. Most of them lead to the proteolysis of the potentially harmful mislocalised, misfolded or damaged mitochondrial proteins inside or outside of the organelle.

In cases where protein import is impaired, proteins do not accumulate in the cytosol but are degraded by mitochondria-associated degradation (MAD) pathways[13,14]. Protein import deficiencies can lead to different stresses which trigger distinct pathways, for example mitochondrial precursor over-accumulation stress (mPOS)[15], mitochondrial compromised protein import response (MitoCPR)[16], mitochondrial ribosomal quality control (mitoRQC)[17], the unfolded protein response activated by mistargeting of proteins (UPRam)[18] and the constitutively active mitochondrial protein translocation-associated degradation (mitoTAD)[19]. In all of these pathways, ubiquitination marks a subset of substrates for degradation by the proteasome in a process catalysed by ubiquitin-activating (E1), ubiquitin-conjugating (E2) and ubiquitin ligase (E3) enzymes. This allows the activation of ubiquitin, and its transfer to a target protein[20,21]. There are many different ubiquitin ligases in the cell which have specificities for distinct substrates, but which, in the case of MAD pathways, have not been identified yet.

MQC pathways involve further cytosolic factors. Efficient import of nuclearly encoded mitochondrial proteins requires a series of cytosolic chaperones, including Hsp70 and Hsp90, which allow correctly translated proteins in the cytosol to be retained in a partially unfolded, non-aggregated, import competent state[22,23]. Thus, a response to the impairment of mitochondrial protein import often includes upregulation of cytosolic chaperones. Ubiquilins also have roles in the degradation of non-imported mitochondrial membrane proteins. These bind to the hydrophobic domains of import substrates in the cytosol preventing their aggregation[24,25]. This allows the substrates to interact with the mitochondrial outer membrane (OM), facilitating their import. However, if an import defect occurs, the substrates remain bound to the ubiquilin and are subsequently diverted to a MAD pathway.

Many MQC pathways involve cytosolic feedback loops with transcriptional control. In yeast, transcription heat shock factor 1 (Hsf1) induces the upregulation of cytosolic chaperones and the proteasome, but attenuates cytosolic translation and the expression of oxidative phosphorylation proteins[26]. Thus, Hsf1 contributes to the mPOS, mitoTAD, UPRam and mitoCPR pathways[15,16,18,19].

In *C. elegans*, mitochondrial proteostasis is regulated by transcription factor ATFS-1 which contains both a mitochondrial targeting signal and a nuclear localisation sequence. ATFS-1 is normally mitochondrially localised. However, upon mitochondrial dysfunction, its import is abolished and it instead is imported into the nucleus[27,28] where it induces increased expression of mitochondrial chaperones, proteases, protein translocases and complex assembly factors, while abundant metabolic mitochondrial proteins are downregulated[28–30]. ATF4/ATF5 and the HAP complex work in analogous processes, termed the mitochondrial unfolded protein response (UPR$^{mt}$), in humans and yeast respectively[26,31,32].

In recent years, MQC processes have become a major focus of research. However, essentially all studies have focused on yeast and metazoans, mainly mammals and *C. elegans*, all of which belong to the eukaryotic supergroup of the Opisthokonts. Eukaryotes however comprise at least five different supergroups, indicating that we do not know to what extent MQC processes are a general feature of eukaryotes[33].

Here we have studied MQC in the protozoan *Trypanosoma brucei*, which belongs to the subgroup Discoba of the paraphyletic Excavate supergroup. *T. brucei* has a complex life cycle, living either in the tsetse fly, where its mitochondrion is capable of oxidative phosphorylation, or in the bloodstream of a mammalian host, where oxidative phosphorylation and other mitochondrial functions are repressed[34]. In the last decade, the *T. brucei* mitochondrion emerged as arguably the best studied such organelle outside yeast and mammals[35]. Due to its evolutionary divergence from Opisthokonts, it has many unique features. The most relevant ones for the study of MQC are that *T. brucei* has a single network-like mitochondrion with a single unit mitochondrial genome[36,37], and that its transcription in the nucleus is exclusively polycistronic[38]. Hence classic mitophagy cannot be used as a MQC pathway since it would destroy the single mitochondrion. Moreover, polycistronic transcription prevents transcriptional regulation, indicating that MQC in trypanosomes is expected to function posttranscriptionally. Finally, other than ubiquitin and the proteasome[39–42], orthologues of most common MQC factors found in yeast and metazoa are absent in *T. brucei*.

The mitochondrial import machineries in *T. brucei* show significant deviations from those of Opisthokonts that have been studied in great detail[43,44]. The protein translocase of the OM (TOM) in *T. brucei* was termed atypical TOM (ATOM)[45]. Besides the import pore ATOM40 which is an unusual Tom40 homolog[46,47] and ATOM14 which is a highly diverged Tom22 homolog[48], the remaining five ATOM subunits are unique to trypanosomatids. These include the two essential protein import receptors, ATOM46 and ATOM69, which have substrate preferences for presequence-containing, more hydrophilic proteins and presequence-lacking, more hydrophobic proteins, respectively[49]. However, while these substrate preferences mirror those of the yeast protein import receptors Tom20 and Tom70, the yeast and trypanosomal receptor pairs are not homologs but arose independently by convergent evolution[50].

In order to investigate MQC in trypanosomes, here in this work, we impair mitochondrial protein import by inducible

ablation of each of the seven essential subunits of the ATOM complex. In contrast to all other subunits, ablation of ATOM69, the import receptor preferring presequence-lacking hydrophobic proteins, results in the mitochondrial recruitment of the proteasome and several trypanosomatid-specific proteins. Some of the latter are essential for proper function of a MQC pathway that is triggered by ablation of ATOM69. One of the identified MQC factors is a nuclear-localised ubiquitin-like protein that, upon import stress, is released into the cytosol in a process essential for this pathway.

## Results

**Ablation of ATOM69 stimulates recruitment of proteins to the mitochondrion.** Using stable isotope labelling with amino acids in cell culture and mass spectrometry (SILAC-MS), we recently analysed the impact of tetracycline-inducible RNAi-mediated ablation of ATOM69 in *T. brucei* to identify the substrates of this central protein import receptor[49]. The results showed that, while, as expected, many mitochondrial proteins are depleted after ablation of ATOM69, a small number of other proteins present in the mitochondrial fraction actually increased in abundance. To investigate whether this is a general response to a mitochondrial protein import defect, we carried out the same experiment for cell lines allowing tetracycline-inducible ablation of the six remaining ATOM subunits, all of which are essential for normal growth[46,50,51] as outlined in Fig. 1a. In our systematic proteomics study of all seven ATOM subunits, we integrated raw data of SILAC-MS analyses of mitochondria-enriched fractions of ATOM40, ATOM69 and ATOM46 RNAi cell lines as previously published[49,52]. Quantitative proteomics data were jointly analysed using a linear model[53,54] and visualised in volcano plots to reveal changes in protein abundance in the crude mitochondrial extracts of all RNAi cell lines following inducible ablation of the target protein (Fig. 1b, c; Supplementary Data 1a). In each RNAi experiment, the targeted ATOM subunit was efficiently downregulated (Fig. 1b, c). In agreement with previous findings[49,52], the ablation of core subunits ATOM40, ATOM19, ATOM14, ATOM12 and ATOM11 results in the downregulation of other ATOM subunits (Fig. 1b), whereas ablation of the receptor subunits ATOM46 and ATOM69 leaves the ATOM complex largely intact[49,50] (Fig. 1c). Moreover, in all datasets, we see a general depletion of mitochondrial proteins (Fig. 1b, c), as might be expected since all ATOM subunits are essential for protein import[50]. Strikingly, the ATOM69 RNAi cell line stands out when looking at proteins with increased abundance in the mitochondrial fractions upon depletion of an ATOM subunit. The volcano plot for this cell line shows by far the highest number of non-mitochondrial proteins enriched in crude mitochondrial fractions of all data sets (Fig. 1).

A few proteins with increased abundance in the mitochondria-enriched fractions are also seen after ablation of ATOM40, ATOM14 and ATOM11. However, ablation of ATOM40 and ATOM11 results in an approximately 3-fold reduction of ATOM69. Thus, the increase in the abundance of non-mitochondrial proteins seen in these two cell lines might be an indirect effect of reduced ATOM69 levels. In case of the ATOM14 RNAi cell line, ATOM69, as with the other ATOM subunits, is only marginally affected, yet the levels of mitochondrial proteins are most strongly reduced in this cell line. This could be because ATOM14, like its orthologue Tom22 in yeast, has a dual function as an assembly factor and a secondary import receptor[48]. The severe effect on mitochondrial protein import may also explain why the levels of some non-mitochondrial proteins are increased in the very efficient ATOM14 RNAi cell line. In the remaining data sets for ATOM19 and ATOM12 (Fig. 1b), as well as for the

other receptor subunit ATOM46 (Fig. 1c), only a small number of enriched proteins are detected in the crude mitochondrial fraction. Based on these findings we propose that ablation of the mitochondrial protein import receptor ATOM69, but not of any of the other ATOM subunits, triggers the recruitment of cytosolic proteins to mitochondria. Notably, depletion of ATOM69 affects in particular proteins of two central, highly abundant mitochondrial machineries, the mitochondrial oxidative phosphorylation system and mitochondrial ribosomes, (Supplementary Fig. 1, cluster 2, 3 and 5, see Supplementary Data 1b), precursors of which can be expected to aggregate when accumulating in the cytosol.

We previously showed that ablation of ATOM40, the main protein translocation pore in the OM of the trypanosomal mitochondrion, results in a growth arrest and subsequent proteasomal degradation of non-imported mitochondrial proteins in the cytosol[52]. Thus, we wondered whether proteasomal removal of non-imported proteins also occurs in ATOM69-ablated cells. To test this, we performed a SILAC-based proteomic analysis of whole cell extracts after ablation of ATOM69 (Fig. 2a). As was observed in the mitochondrial fractions, the vast majority of mitochondrial proteins are depleted in this dataset, indicating that, as in case of ATOM40 depletion, these proteins are likely degraded by the proteasome when their import is inhibited (Fig. 2a, top). Interestingly, while the depletion of some non-mitochondrial proteins is also seen (Fig. 2a, bottom), mitochondrial proteins are clearly the preferred substrates for the cytosolic degradation pathway that is induced by ablation of ATOM69. Furthermore, in contrast to what is observed in the mitochondrial fractions (Fig. 1c, right panel), essentially no upregulation of proteins could be observed in this whole cell sample. Hence, the steady state levels of these proteins do not change upon ablation of ATOM69. This suggests that these proteins are recruited to the mitochondrion under these conditions. To test this, we established ratio-ratio plots for known mitochondrial proteins and the other remaining proteins quantified in whole cell extracts and mitochondria-enriched fractions from ATOM69-ablated cells. As expected, the levels of mitochondrial proteins are largely equally decreased in both fractions (Fig. 2b, top), whereas many non-mitochondrial proteins are clearly enriched in crude mitochondrial fractions (Fig. 2b, bottom). In other organisms, initiation of mitochondrial quality control pathways often results in the specific recruitment of proteins to mitochondria[55,56]. Could it be that some of these recruited proteins are involved in a cytosolic trypanosomal MQC pathway that ultimately leads to the cytosolic degradation of non-imported proteins after ablation of ATOM69?

To address this question, we selected four putative cytosolic candidates (Tb927.10.2290, Tb927.11.4130, Tb927.8.1590, Tb927.9.7200) from the large pool of recruited proteins for further investigation. For selection, we also considered our previously published SILAC-MS data of the ATOM69 RNAi cell line[49], applying the following criteria: >3-fold enriched; *p*-value < 0.05; MaxQuant score >10. From the five remaining proteins, we excluded the putative nucleolar protein UTP14. All four proteins were also among the top seven most enriched proteins in our new extended analysis of changes in protein levels in mitochondrial fractions versus whole cell extracts of the ATOM69-ablated cells (Fig. 2b, bottom).

Tb927.9.7200 was also enriched in the mitochondrial fractions of ATOM11-depleted cells (Supplementary Fig. 2a, bottom panel). A similar situation is seen for Tb927.10.2290, which is also recruited to mitochondria in ATOM40 and ATOM14-depleted cells, and for Tb927.11.4130 which is enriched in ATOM14-depleted mitochondria, albeit in the case of Tb927.11.4130 to a lesser extent than in the ATOM69 RNAi

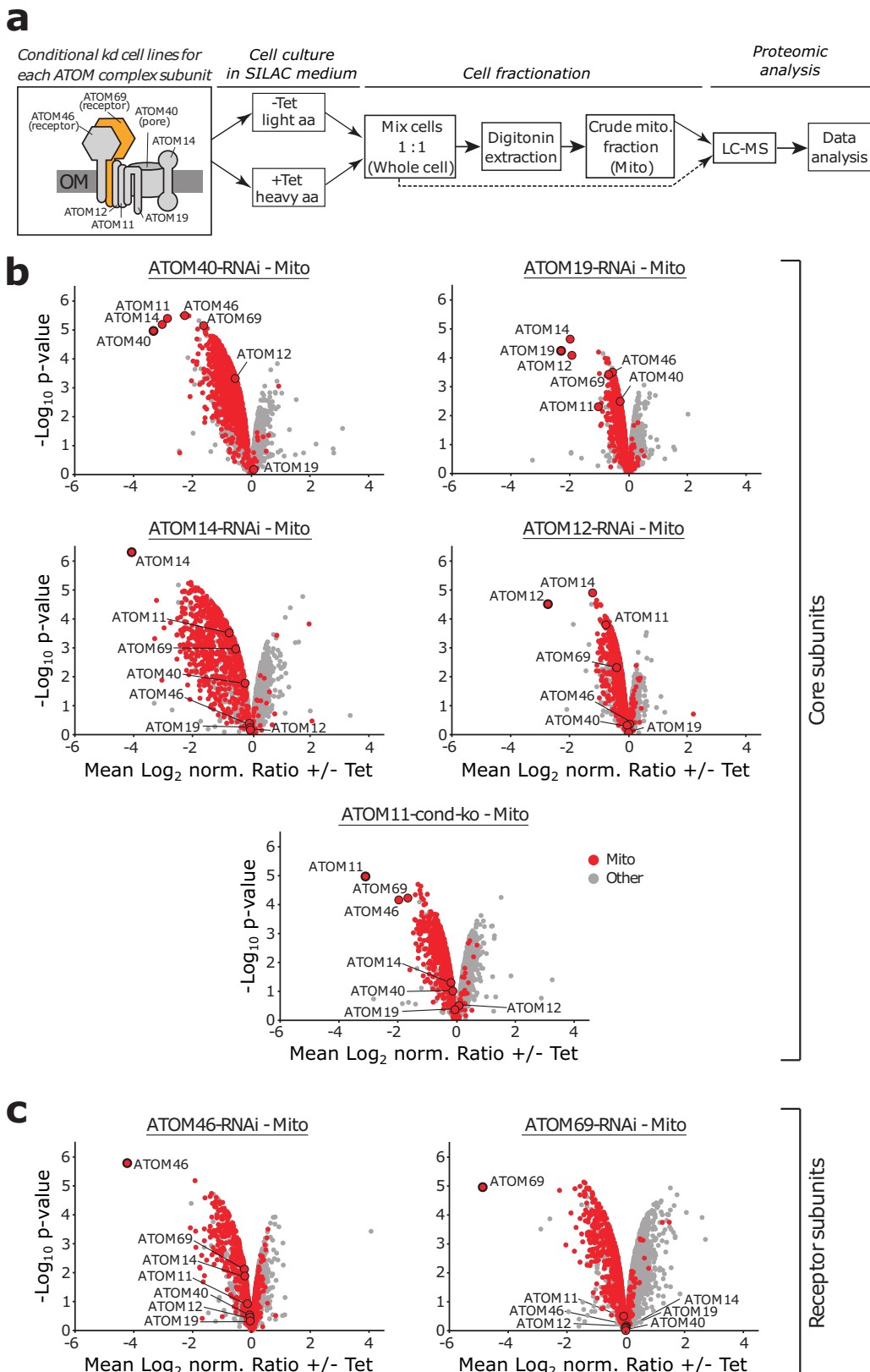

cell line (Supplementary Fig. 2a and Supplementary Fig. 2b). These observations can be explained by the fact that, as discussed above, depletion of ATOM11 and ATOM40 also considerably reduce the levels of ATOM69 (Fig. 1b) and because the highly efficient ablation of ATOM14 causes the most dramatic protein import phenotype of all tested cell lines.

Thus, the mitochondrial recruitment of the four selected proteins is not a general response to the lack of an ATOM complex subunit but appears to be specifically linked to a reduced level of ATOM69. This specificity can best be illustrated by comparing the ATOM69 and the ATOM46-RNAi cell lines (Fig. 1c and Supplementary Fig. 2b). ATOM69 and ATOM46 are both protein import receptors

**Fig. 1 Mitochondrial proteome changes after ablation of ATOM complex subunits. a** General workflow of the quantitative proteomic analysis. The box on the left indicates a schematic model of the atypical translocase of the outer membrane (ATOM) complex with its seven subunits. Receptor subunits and the protein-conducting pore are indicated. ATOM69 is highlighted in orange. Using the stable isotope labelling by amino acids in cell culture (SILAC) method, six inducible RNAi cell lines and a conditional knock out cell line (ATOM11) were differentially labelled with stable-isotope coded heavy arginine (Arg10) and lysine (Lys8) or their light counterparts (Arg0/Lys0). Subsequently, induced (+Tet) and uninduced (−Tet) cells were mixed and crude mitochondrial fractions were purified by digitonin extractions ($n = 3$). For the ATOM69 RNAi cell line, whole cell samples were also analysed (see Fig. 2). All samples were subjected to liquid chromatography-mass spectrometry (LC-MS) followed by computational data analysis. **b, c** Volcano plots visualising SILAC-based quantitative MS data of crude mitochondrial extracts (Mito) for the ATOM core subunits ATOM40, ATOM19, ATOM14, ATOM12 and ATOM11 (**b**) and the ATOM receptor subunits ATOM46 and ATOM69 (**c**). Induction times for the indicated cell lines grown in SILAC medium were ATOM40 (3 d), ATOM19 (3.17 d), ATOM14 (3 d), ATOM12 (3 d), ATOM11 (4 d), ATOM46 (6 d) and ATOM69 (5 d).

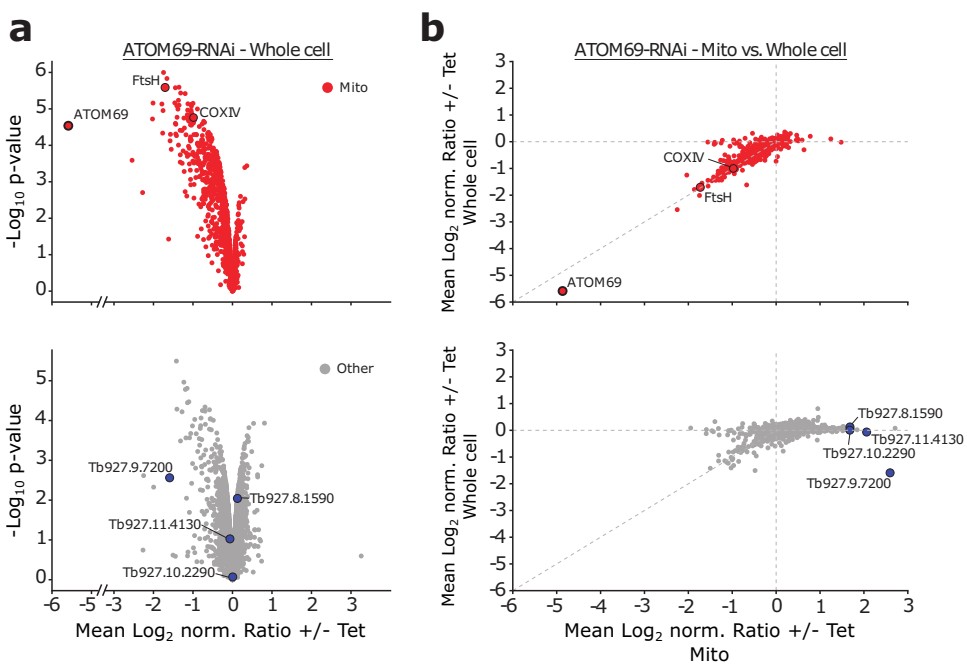

**Fig. 2 Ablation of ATOM69 stimulates recruitment of proteins to the mitochondrion. a** Whole cellular extracts of uninduced and induced ATOM69-RNAi cells were subjected to stable isotope labelling by amino acids in cell culture (SILAC)-based quantitative mass spectrometry (MS). Top, volcano plot depicting mitochondrial proteins (red dots) as defined in[52]. The model substrates FtsH and cytochrome oxidase subunit IV (COXIV) are highlighted. Bottom, volcano plot depicting other proteins (grey dots). Highlighted in blue are the four putative non-mitochondrial candidate proteins that were further studied. **b** Ratio plots depicting changes in protein levels in crude mitochondrial fractions (Mito) *versus* whole cell extracts of ATOM69 RNAi experiments. Proteins are labelled as in **a**.

which are downregulated to large extents in their corresponding RNAi cell lines (approximately 30- and 20-fold respectively). Additionally, their ablation does not affect the level of any other ATOM subunit. However, whereas ablation of ATOM46 failed to recruit additional proteins to the crude mitochondrial fraction, very efficient recruitment of proteins including our four selected candidates is observed after ablation of ATOM69.

The levels of the selected proteins in whole cells did not change after ablation of ATOM69, with the exception of Tb927.9.7200, which had its level reduced (Fig. 2b, bottom panel). Interestingly, while the mitochondrial levels of Tb927.9.7200 strongly increased after ablation of ATOM69, ablation of the protein conduction channel ATOM40 results in a strong decrease in the mitochondrial levels of this protein (Supplementary Fig. 2). In line with these results, immunofluorescence (IF) analysis revealed that C-terminally tagged Tb927.9.7200 localises to the mitochondrion in non-stressed cells (Supplementary Fig. 3). In a similar fashion to our four candidate proteins, although to a lesser extent, recruitment of many subunits of the proteasome (Supplementary Fig. 4a), cytosolic translation factors (Supplementary Fig. 4b) and cytosolic ribosomal proteins was also observed (Supplementary Fig. 4c).

**Three recruited proteins have a role in the degradation pathway.** We surmised that the four selected proteins recruited to mitochondria are promising candidates for involvement in the postulated pathway that removes non-imported proteins after ATOM69 depletion. To investigate whether this indeed is the case, we selected the mitochondrial protein FtsH (Tb927.11.14730, also termed FtsH14[57]) as a model substrate. FtsH is among the most strongly downregulated proteins upon ATOM69 depletion (highlighted in Fig. 2) whereas its level remains essentially unchanged after depletion of ATOM46. In line with this data, biochemical binding assays using the soluble domains of the ATOM69 and ATOM46 receptors showed that FtsH preferentially binds to ATOM69[49]. FtsH is therefore most likely a preferred substrate for the cytosolic degradation pathway triggered by the ablation of ATOM69. Thus, FtsH was in situ HA-tagged at its C-terminus in the background of the ATOM69 RNAi cell line. The immunoblot in Fig. 3a (left panel) shows that inducing ATOM69 RNAi for 3 days caused a 4-fold reduction in the levels of FtsH-HA (Fig. 3b), in agreement with the SILAC RNAi data (Fig. 2a). Moreover, inhibiting the proteasome by MG132 treatment partially restored FtsH-HA levels, suggesting that, as expected, the reduction in the

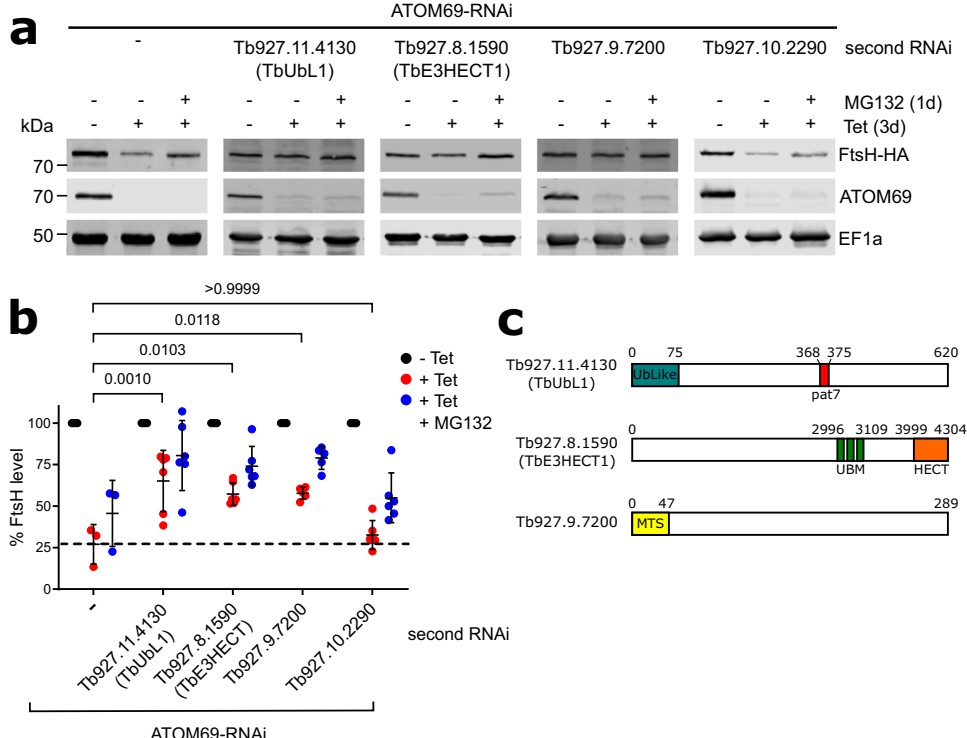

**Fig. 3 Three recruited proteins have a role in mitochondrial quality control (MQC). a** Immunoblots depicting the change in abundance of a 3x HA-tagged model substrate FtsH (FtsH-HA) in whole cells. RNAi-mediated ablation of atypical translocase of the outer membrane 69 (ATOM69) by tetracycline (−/+ tet) was either performed alone (left panel), or in combination with RNAi-mediated ablation of each of the four candidate proteins (right panels), in the absence and presence of MG132 (1d, 500 nM). $2 \times 10^6$ cells were loaded per lane. elongation factor 1a (EF1a) is used as a loading control. **b** Densitometric quantification of the immunoblot signals of FtsH-HA in the RNAi cell lines shown in **a**. The signal in the uninduced RNAi cell lines (-Tet) was set to 100%. The level of FtsH-HA for each sample was normalised to its respective EF1a signal. Data are presented as mean values with error bars corresponding to the standard deviation of six independent biological replicates, except for the ATOM69-RNAi cell where three replicates were used. Statistical analysis was performed using a one-way ANOVA followed by a Bonferroni posthoc test to allow for multiple comparisons. The *p* values calculated are as indicated in the graph. **c** Predicted domain structure of the three candidate proteins that contribute to the MQC pathway. Tb927.11.4130, termed ubiquitin-like protein 1 (TbUbL1), contains an Ub-Like domain (blue) and a Pat7 monopartite NLS sequence (red). Tb927.8.1590, termed TbE3HECT1, contains three ubiquitin-binding motifs (green) and an E3 ligase C-terminal HECT domain (orange). Tb927.9.7200 contains a predicted N-terminal mitochondrial targeting signal (yellow). Source data are provided as a Source Data file.

levels of FtsH after ablation of ATOM69 might be due to proteasome activity.

To show whether the four candidate proteins are involved in the degradation of FtsH-HA induced by the lack of ATOM69, we generated a series of RNAi cell lines where both ATOM69 and each candidate individually were targeted for RNAi simultaneously. In addition, these cell lines also expressed the in situ tagged FtsH-HA protein. As a control, individual RNAi cell lines for the four candidate proteins were also generated. None of these single RNAi cell lines showed a growth phenotype, with the exception of Tb927.8.1590, where its ablation marginally slowed down growth (Supplementary Fig. 5, top panels). For the four double RNAi cell lines, on the other hand, the same growth phenotype is seen as for the parent ATOM69 RNAi cell line: a growth arrest starting around four days after induction[50] (Supplementary Fig. 5, bottom panels). Immunoblots were then used to assess the levels of FtsH-HA in the four double RNAi cell lines (Fig. 3a). The results showed that ablation of Tb927.11.4130, Tb927.8.1590 and Tb927.9.7200 in parallel with ATOM69 partially restored the levels of FtsH-HA (Fig. 3a). The quantification of triplicate experiments demonstrates that the increase in the FtsH levels was significant when compared to ATOM69 depletion alone (Fig. 3b). However, the same was not the case for Tb927.10.2290, the trypanosomal homologue of yeast Ydj1. This suggests that Tb927.11.4130, Tb927.8.1590 and Tb927.9.7200, but not Tb927.10.2290, function in a MQC pathway

induced by the ablation of ATOM69, and that in their absence FtsH-HA cannot efficiently be degraded by the proteasome.

Interestingly, all three identified putative MQC factors are kinetoplastid-specific (Fig. 3c). The first candidate, Tb927.11.4130 (MW: 67 kDa), contains an N-terminal ubiquitin-like domain and a central nuclear localisation sequence (NLS, pat7). It was termed TbUbL1. The ubiquitin-like domain (UBL) of TbUbL1, like the corresponding domain in human Parkin and yeast Dsk2, is diverged from the sequence of ubiquitin itself (Supplementary Fig. 6a), but contains the beta-grasp fold of ubiquitin with a conserved proteasome-interacting motif (PIM) (Supplementary Fig. 6b). The second factor, Tb927.8.1590, is a very large protein of 470 kDa; it contains a C-terminal HECT E3 ubiquitin ligase domain, along with three ubiquitin binding motifs (UBM) in the C-terminal half of the protein, and was termed TbE3HECT1. The third candidate, Tb927.9.7200, 34 kDa, is a trypanosomatid-specific protein of unknown function that interestingly has a predicted N-terminal mitochondrial targeting sequence (MTS).

**Import defect causes a release of nuclear TbUbL1 to the cytosol.** We decided to analyse TbUbL1 in more detail as it contains a UBL domain found in MQC factors of other organisms. As TbUbL1 also contains a predicted nuclear localisation signal, we investigated its localisation. To that end, TbUbL1 was myc-tagged

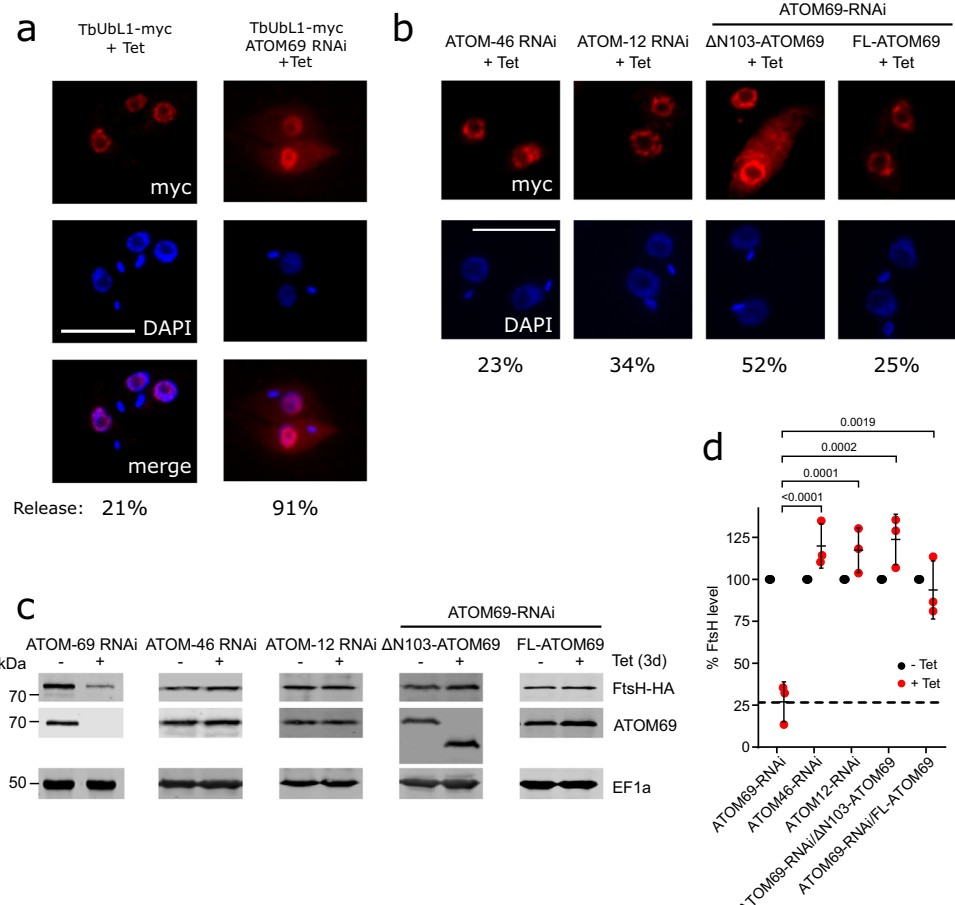

**Fig. 4 Import defect induced by ATOM69 RNAi but not by other ATOM subunits causes a release of nuclear TbUbL1 to the cytosol. a** Left, immunofluorescence (IF) analysis of a cell line allowing tetracycline-inducible expression of myc-tagged ubiquitin-like protein 1 (TbUbL1-myc +Tet). Right, IF analysis of a cell line allowing tetracycline-inducible expression of myc-tagged TbUbL1 together with simultaneous RNAi of the atypical translocase of the outer membrane 69 (ATOM69) (TbUbL1-myc/ATOM69-RNAi +Tet). Scale bar 10 μm. Cells were induced for 3d. The % of the population displaying a TbUbL1-myc nuclear release phenotype is recorded below the images, 100 cells of each were analysed. This IF analysis was repeated independently at least three times with similar results. **b** IF analysis of cells expressing TbUbL1-myc with simultaneous RNAi of ATOM46, ATOM12 or ATOM69. Scale bar 10 μm. Cells were induced for 2d (cell lines based on ATOM12 RNAi) or 3d (cell lines based on ATOM46 or ATOM69 RNAi). The ATOM69 RNAi cell lines were combined with tetracycline-inducible expression of either N-terminally truncated (ΔN103-) or full length (FL-) ATOM69, respectively. This IF analysis was repeated independently at least two times with similar results. **c** Immunoblots depicting the change in abundance of the model substrate FtsH-HA in whole cells after tetracycline-inducible RNAi-mediated ablation of ATOM69, ATOM46 or ATOM12, or of ATOM69 in combination with tetracycline-inducible expression of either N-terminally truncated (ΔN103-) or full length (FL-) ATOM69 as indicated. Note that for the ATOM69 RNAi cell line, the sections of blots used are already shown in Fig. 2a. $2 \times 10^6$ cells were loaded per lane. Elongation factor 1a (EF1a) is used as a loading control. **d** Quantification of triplicate immunoblots shown in **c**, with the FtsH-HA level of each uninduced cell line set to 100%. The level of FtsH-HA for each sample was normalised to its respective EF1a signal. Data are presented as mean values with error bars corresponding to the standard deviation of the mean of three independent biological replicates. Significance was calculated using a one way ANOVA followed by a Bonferroni posthoc test to allow for multiple comparisons, with the $p$ values calculated indicated in the graph. Source data are provided as a Source Data file.

at its C-terminus, and ectopically expressed under tetracycline control. IF microscopy analysis of tetracycline-induced cells using anti-myc antibodies indicates that TbUbL1 indeed localises to foci at the periphery of the nucleus in the majority of cells (Fig. 4a). However, upon the induction of ATOM69 RNAi, TbUbL1-myc was detected over the whole nucleus with the exception of the nucleolus, and interestingly, in approximately 91% of cells a fraction of the protein was released into the cytosol. For a quantification of the TbUbL1 release from the nucleus see Supplementary Fig. 7a. The level of TbUbL1-myc expressed was unchanged whether or not ATOM69 was depleted by RNAi (Supplementary Fig. 8a, right panel). The same change in localisation of TbUbL1 could also be induced by addition of the membrane uncoupler carbonylcyanide-m-chlorophenylhydrazone (CCCP) (Supplementary Fig. 8a), or by ATOM40 RNAi (Supplementary Fig. 8b). Both

of these treatments are expected to decrease mitochondrial protein import, but do not affect the levels of TbUbL1-myc (Supplementary Fig. 8a, b, right panels). However, as ablation of ATOM40 reduces the levels of ATOM69 (Fig. 1), the release of TbUbL1 under these conditions is likely an indirect effect. Notably, release of TbUbL1 into the cytosol is not a general stress response to RNAi, since neither RNAi-mediated ablation of the mitochondrial ribosomal protein MRPS5[58], nor of KMP11, a protein involved in cytokinesis[59], had an impact on the nuclear localisation or the intracellular levels of TbUbL1-myc (Supplementary Fig. 8b). Hence, nuclear release of TbUbL1 is a hallmark of the MQC pathway induced by ablation of ATOM69.

In order to investigate how tightly the release of TbUbL1 is connected to the absence of ATOM69, we tested whether it also occurs in ATOM46 and ATOM12 RNAi cell lines. We chose the

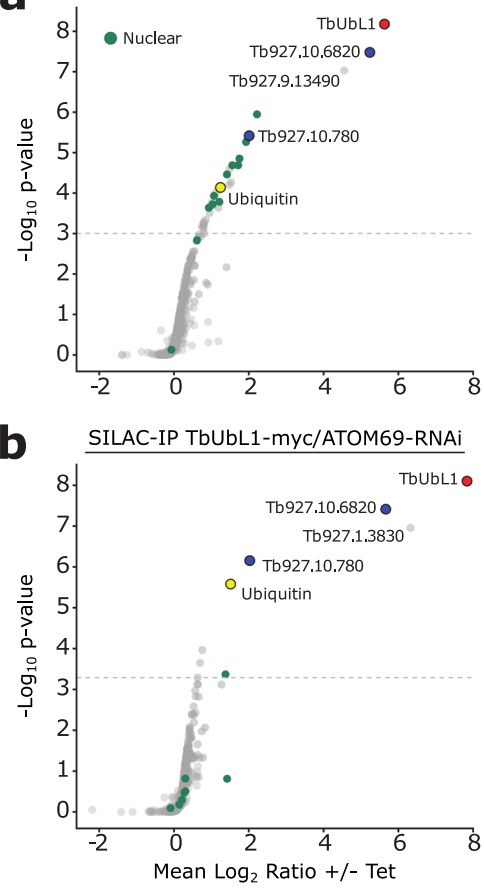

**Fig. 5 SILAC-IP of TbUbL1-myc from whole cell extracts in the presence and absence of ATOM69.** Stable isotope labelling by amino acids in cell culture-immunoprecipitation (SILAC-IP) experiments of **a** wildtype cells (TbUbL1-myc +Tet) or **b** an induced atypical translocase of the outer membrane 69 (ATOM69) RNAi cell line (TbUbL1-myc/ATOM69-RNAi +Tet) both expressing tetracycline inducible ubiquitin-like protein 1 (TbUbL1). Proteins shown were quantified in 3/3 independent replicates. *p* values were determined following the rank sum method[83]. Highlighted are the bait TbUbL1-myc (red), ubiquitin (yellow), two other proteins that are enriched in both cell lines (blue), and nuclear proteins (green). Annotation of nuclear proteins is based on Tryptag[87] or the nuclear proteome[88]. The horizontal dashed lines indicate a false discovery rate of 5%.

ATOM46 receptor because, in contrast to ATOM69, it prefers hydrophilic presequence-containing substrates. Moreover, in RNAi cell lines targeting ATOM46 or ATOM12, the levels of ATOM69 are not affected (Fig. 1b, c). The IF in Fig. 4b shows that populations of ATOM46 and ATOM12 ablated cells exhibit a TbUbL1 nuclear release of 23% and 34% respectively, which is similar to the 21% release that is observed in cells where no RNAi is induced (Fig. 4a). In the population of ATOM69 depleted cells, in comparison, 91% of cells show release of TbUbL1 from the nucleus (Fig. 4a). Moreover, TbUbL1-myc protein is expressed to an equal extent in these RNAi cells (Supplementary Fig. 9a). Thus, cytosolic accumulation of TbUbL1 appears to be a specific response to the absence of ATOM69 and does not occur to the same extent after ablation of ATOM46 and ATOM12, respectively. In line with this claim, FtsH-HA levels were not affected upon ATOM46 or ATOM12 depletion (Fig. 4c, d).

RNAi-mediated ablation of ATOM69 is a harsh treatment from which cells cannot recover. We were therefore wondering whether nuclear release of TbUbL1 can be triggered by a more gentle

perturbation. Intriguingly, ATOM69 contains an N-terminal CS/Hsp20-like chaperone-binding domain. Previous work has shown that removal of this domain only marginally slows down growth under standard conditions but does not kill the cells[49]. Thus, we produced a cell line that exclusively expresses an untagged ATOM69 variant lacking the CS/Hsp20-like domain (ΔN103-ATOM69)[49], along with a myc-tagged version of TbUbL1. We also produced an analogous cell line exclusively expressing the full length untagged version of ATOM69 as a control (FL-ATOM69). The IF analysis in Fig. 4b shows that expression of the ATOM69 variant lacking the CS/Hsp20-like domain was sufficient to induce nuclear release of tagged TbUbL1 in 52% of the cells, whereas expression of the full length version caused a release in only 25% of all cells. However, the levels of FtsH-HA were not affected in either of the two cell lines (Fig. 4c, d). This is in line with the marginal growth phenotype observed in the cell line expressing ΔN103-ATOM69 (Supplementary Fig. 9b). TbUbL1-myc protein was expressed to an equal extent in all of these cell lines (Supplementary Fig. 9a). The simplest explanation for these results is that, in the absence of the ATOM69 CS/Hsp20-like domain, a small amount of non-imported proteins aggregates in the cytosol. This would trigger the observed nuclear release of TbUbL1 and result in the degradation of the aggregates by the proteasome, allowing the cell to keep proliferating, albeit at a reduced rate.

In summary, these results show that the extensive release of TbUbL1 to the cytosol is associated with the lack of ATOM69 and is not observed after ablation of ATOM46 or ATOM12. Moreover, triggering of the pathway does not require removal of the whole receptor: the absence of the CS/Hsp20-like domain from ATOM69 alone is sufficient to trigger nuclear release of TbUbL1.

Next, we performed SILAC-immunoprecipitation (IP) analyses in WT and ATOM69 ablated cells using tetracycline-induced TbUbL1-myc as a bait. Under both conditions, TbUbL1-myc precipitated the same three proteins with enrichment factors of more than 10-fold: Tb927.10.6820, a trypanosome-specific protein of unknown function and Tb927.10.780, a putative RING domain containing E3 ubiquitin ligase, with ubiquitin itself also recovered although to a lesser extent (Fig. 5; Supplementary Data 2). Under WT conditions, 12 nuclear proteins, including all five subunits of replication factor C, were significantly enriched more than two-fold (Fig. 5a), whereas after ablation of ATOM69 only a single nuclear protein was significantly enriched in the eluate (Fig. 5b, green dots). In addition to these proteins, Tb927.9.13490, an as-yet uncharacterised metallopeptidase, was detected in the presence of ATOM69; however since peptidases of this type have been recovered in unrelated SILAC-IPs, we considered it to be a contaminant. Moreover, in the absence of ATOM69, glucose-6-phosphate isomerase (Tb927.1.3830) was detected. The relevance of this finding is not clear since in trypanosomes this enzyme is localised in the glycosomes. In summary, the results of the SILAC-IP analysis are in line with the IF analysis shown in Fig. 4a. They confirm the nuclear localization of TbUbL1 in wild-type cells, as well as its release into the cytosol after ablation of ATOM69.

**Nuclear export of TbUbL1 is essential for proteasomal degradation of FtsH.** TbUbL1 is involved in the proteasomal degradation of mitochondrial proteins that accumulate in the cytosol after ablation of ATOM69 (Fig. 3b). To find out whether this function is linked to the release of TbUbL1 from the nucleus that is triggered by a mitochondrial protein import defect, we used the nuclear export inhibitor leptomycin B (LMB). LMB inhibits the CRM1-dependent pathway of nuclear protein export[60], and was previously shown to function in this way in trypanosomes[61]. The IF microscopy analysis in Fig. 6a demonstrates that LMB

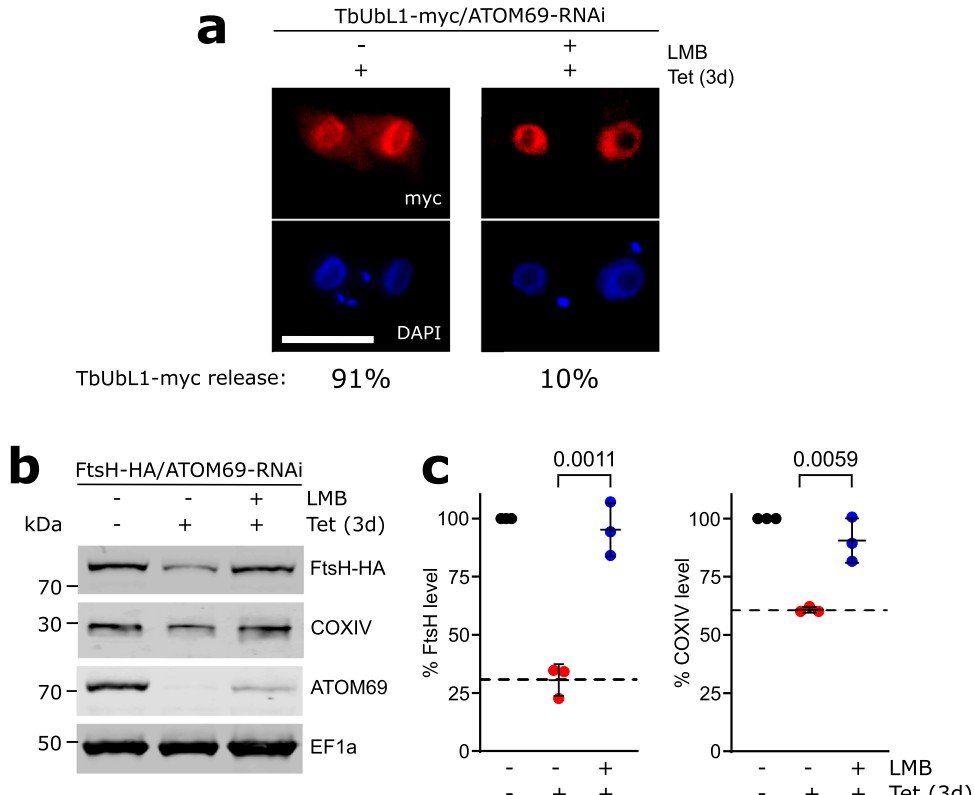

**Fig. 6 Inhibition of nuclear protein export interferes with mitochondrial quality control (MQC). a** IF analysis of cells expressing ubiquitin-like protein 1 (TbUbL1-myc) with simultaneous atypical translocase of the outer membrane 69 (ATOM69)-RNAi, induced for 3d (left), and additionally treated with 50 ng/ml leptomycin B (LMB) for 1d (right). Scale bar 10 μm. The % of the population displaying a TbUbL1-myc nuclear release phenotype is recorded below the images, 100 cells of each were analysed. This IF analysis was repeated independently at least three times with similar results. **b** Immunoblots depicting the change in abundance of the substrates in situ tagged FtsH-HA and endogenous cytochrome oxidase subunit IV (COXIV) in whole cells after tetracycline-inducible RNAi mediated ablation of ATOM69 in the absence and presence of 50 ng/ml LMB. $2 \times 10^6$ cells were loaded per lane. Elongation factor 1a (EF1a) is used as a loading control. **c** Quantifications of the FtsH-HA and COXIV levels of triplicate immunoblots shown in **b**, with the tagged protein level of each uninduced cell line set to 100%. The level of tagged protein for each sample was normalised to its respective EF1a signal. Data are presented as mean values with error bars corresponding to the standard deviation of the mean of three independent biological replicates. Significance calculated using a two-tailed unpaired $t$-test, with $p$-values indicated. Source data are provided as a Source Data file.

addition, as expected, prevented the release of TbUbL1-myc from the nucleus, provided that mitochondrial protein import was inhibited by ATOM69 RNAi (Fig. 6a). Most importantly, immunoblotting showed that LMB treatment prevented proteasomal degradation of FtsH-HA, despite the presence of an import defect caused by ablation of ATOM69 (Fig. 6b, c). However, while the RNAi was still operational in LMB-treated cells, its efficiency was slightly reduced.

We used LMB-treatment in combination with SILAC-MS to identify the substrate spectrum of TbUbL1. The proteins that were significantly enriched in crude mitochondrial fractions depleted in ATOM69 in the presence of LMB are highlighted in the top right quadrant of the volcano plot (Supplementary Fig. 10a; Supplementary Data 3). The population consists of 109 proteins, of which 59 are mitochondrial proteins, and most of these were efficiently downregulated in LMB-untreated crude mitochondrial fractions lacking ATOM69 (Supplementary Fig. 10b; Supplementary Data 3), suggesting that they are substrates of TbUbL1-mediated degradation in ATOM69 depleted cells. Since ATOM69 was depleted to the same extent in untreated and LMB-treated cells, differential RNAi efficiency can be excluded as a possible confounding factor.

The set of putative substrates of TbUbL1 includes many subunits of respiratory complexes and of the mitoribosome (Supplementary Data S3), and interestingly a number of these

proteins do not contain transmembrane domains, for example cytochrome oxidase subunit IV (COXIV). We verified whether COXIV was a TbUbL1 substrate via western blot (Fig. 6b, c); just like FtsH, COXIV level is reduced in the absence of ATOM69 but restored after LMB-treatment. Whereas import of FtsH depends almost exclusively on ATOM69, the import of COXIV depends equally on both ATOM69 (50.3%) and ATOM46 (49.7%)[49], despite having a mapped N-terminal mitochondrial targeting sequence[62]. However, the majority of COXIV is present in the insoluble pellet along with integral membrane proteins after an alkaline carbonate extraction (Supplementary Fig. 11a). Moreover, when using a recently published aggregation assay[63], both untagged and tagged COXIV are more aggregation prone than two integral membrane and two soluble proteins that serve as controls (Supplementary Fig. 11b), which may explain why the import of COXIV depends 50% on ATOM69. Thus, our SILAC-based analysis greatly expands the number and characteristics of TbUbL1 substrate candidates.

In a second approach to investigate the impact of TbUbL1 release from the nucleus, we placed the previously characterised nuclear localization signal (NLS) of the *T. brucei* La protein[64] at the C-terminus of TbUbL1, in front of the myc tag. NLS sequences are recognised by soluble carrier proteins of the importin-β family, allowing active transport of the NLS-containing protein as cargo. The resulting protein, termed

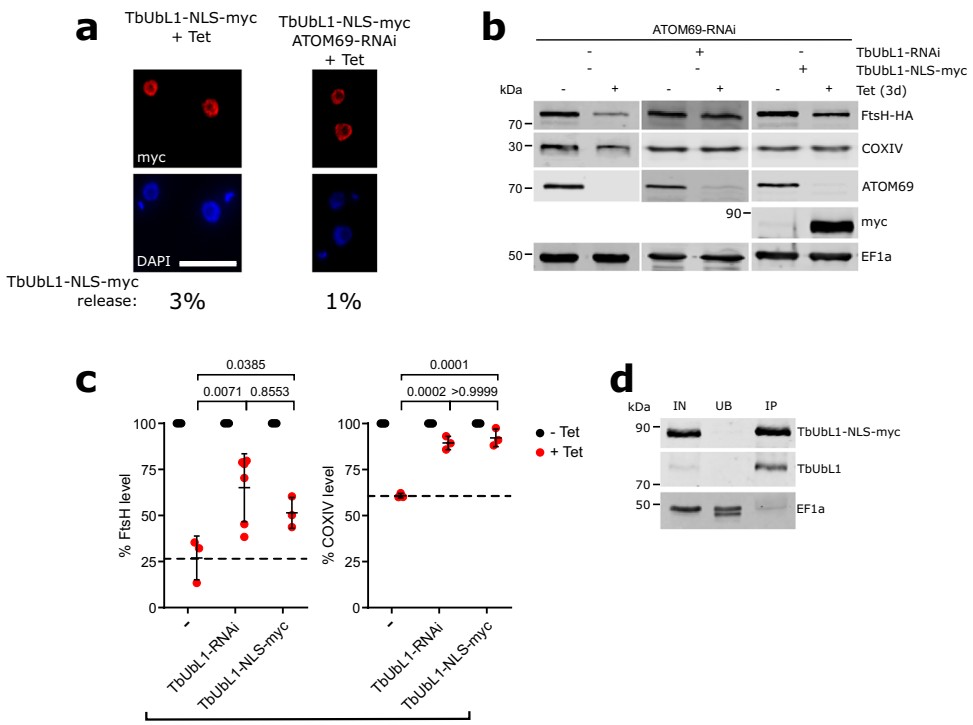

**Fig. 7 Nuclear export of TbUbL1 is essential for proteasomal digestion of FtsH. a** Left, immunofluorescence (IF) analysis of a cell line allowing tetracycline-inducible expression of ubiquitin-like protein 1 (TbUbL1) containing a nuclear localisation signal (NLS) in front of the C-terminal myc-tag (TbUbL1-NLS-myc +Tet). Right panels, IF analysis of a cell line allowing tetracycline-inducible expression of TbUbL1 containing a NLS in front of the C-terminal myc-tag together with simultaneous RNAi of atypical translocase of the outer membrane 69 (ATOM69) (TbUbL1-NLS-myc/ATOM69-RNAi +Tet). Cells were induced for 3d. Scale bar 10 μm. The % of the population displaying a TbUb1-myc nuclear release phenotype is recorded below the images, 100 cells of each were analysed. This IF analysis was repeated independently at least three times with similar results. **b** Immunoblots depicting the change in abundance of in situ tagged FtsH-HA and endogenous cytochrome oxidase subunit IV (COXIV) in whole cells after tetracycline-inducible RNAi-mediated ablation of ATOM69 alone (left panels), in combination with RNAi-mediated ablation of TbUbL1 (middle panel), or in combination with tetracycline-inducible expression of TbUbL1-NLS-myc (right panel) are shown. Note that for the first two cell lines, sections of blots already shown in Fig. 2a, b were used, with the anti-COXIV stained panels shown now in addition. $2 \times 10^6$ cells were loaded per lane. Elongation factor 1a (EF1a) is used as a loading control. **c** Quantification of triplicate immunoblots shown in **b**, with the tagged protein level of each uninduced cell line set to 100%. The level of tagged protein for each sample was normalised to its respective EF1a signal. Data are presented as mean values with error bars corresponding to the standard deviation of the mean of three independent biological replicates, except for the ATOM69/TbUbL1-RNAi cell line where six replicates were used to assess FtsH-HA level, as also used in Fig. 3b. Significance was calculated using a one way ANOVA followed by a Bonferroni posthoc test to allow for multiple comparisons, with the p-values calculated indicated in the graph. **d** Immunoprecipitation using myc-conjugated beads with whole cell extracts of the TbUbL1-NLS-myc expressing cell line induced for 1d. 5% whole cell equivalent (IN), 5% of the unbound protein fraction (UB) and 50% of the final eluate (IP) were separated by SDS-PAGE. The resulting immunoblots were probed with anti-myc-tag antibodies (top panel), antisera against TbUbL1 (middle panel), and EF1a (bottom panel) which is used as a negative control. This IP analysis was repeated independently twice with similar results. Source data are provided as a Source Data file.

TbUbL1-NLS-myc, was ectopically expressed under tetracycline control. The protein could be efficiently expressed but the population of cells showing release of nuclear TbUbL1 was between 1–3% in uninduced and induced cells (Fig. 7a) which is even lower than 21% of cells that showed cytosolic staining of TbUbL1 in the uninduced ATOM69 RNAi cell line (Fig. 4a). For a quantification of the nuclear signals see Supplementary Fig. 7b.

Immunoblotting revealed that expression of TbUbL1-NLS-myc in ATOM69-ablated cells partially restored the levels of FtsH-HA and COXIV in comparison to the parent cell line which only expressed endogenous TbUbL1 (Fig. 7b). Quantification of triplicate experiments showed that these differences were significant (Fig. 7c). Interestingly, the restoration of these levels occurred despite endogenous TbUbL1 still being present. The IP in Fig. 7d demonstrates that the overexpressed TbUbL1-NLS-myc interacts with endogenous TbUbL1 and therefore may prevent nuclear export of the endogenous protein, explaining the dominant negative phenotype.

**Import deficient COXIV is ubiquitinated and interacts with TbUbL1.** After its release from the nucleus, TbUbL1 is recovered in a complex that contains ubiquitin, suggesting it might bind ubiquitinated substrate proteins and possibly hand them over to the proteasome. Thus, we wanted to test whether proteins that have their levels reduced by proteasomal degradation after ablation of ATOM69 can bind to TbUbL1. To that end, we chose the substrate COXIV which has a N-terminal mitochondrial targeting sequence[62] (Fig. 8a, left panel). C-terminally myc-tagged COXIV (COXIV-myc) and a truncated version that lacks the mitochondrial targeting sequence (ΔMTS-COXIV-myc) were expressed in the background of an inducible ATOM69 RNAi cell line. We surmised that, as ΔMTS-COXIV-myc cannot be imported into the mitochondria, use of this truncated protein could facilitate the identification of intermediates of the degradation pathway investigated in this study. The level of the full length COXIV-myc is approximately four-fold reduced after ablation of ATOM69 (Fig. 8a). Interestingly, under the same conditions, the level of

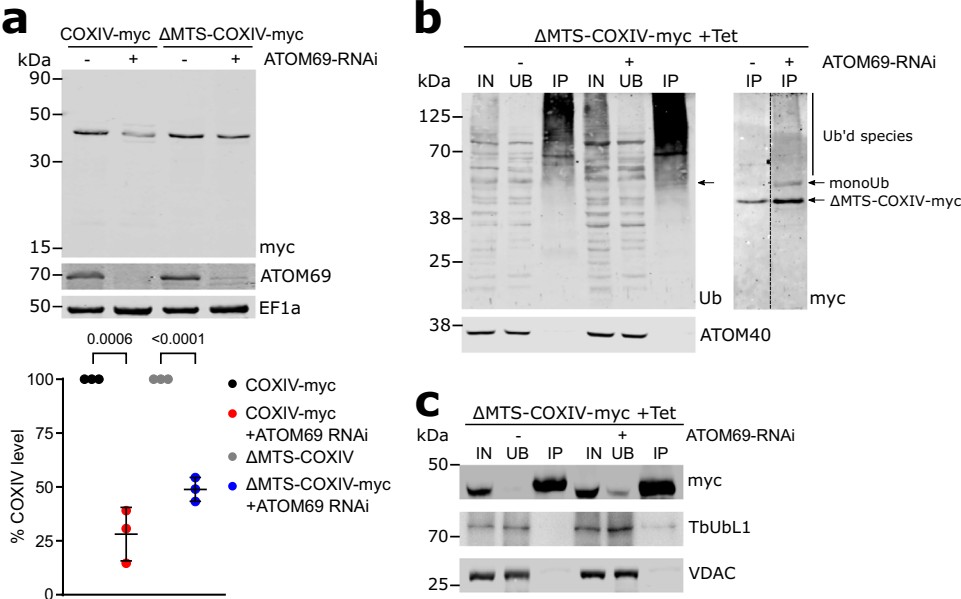

**Fig. 8 Import deficient COXIV is ubiquitinated and interacts with TbUbL1. a** Top panel, immunoblots of whole cell extracts of uninduced and induced atypical translocase of the outer membrane 69 (ATOM69) RNAi cell lines that simultaneously express c-terminally myc-tagged cytochrome oxidase subunit IV (COXIV-myc) or a derivative thereof that lacks the mitochondrial targeting sequence (ΔMTS-COXIV-myc). The immunoblot was probed for the myc-tagged COXIV variants, ATOM69 and elongation factor 1a (EF1a), which serves as a loading control. Bottom panel, quantifications of the levels of the myc-tagged COXIV variants from triplicate immunoblots shown in the top panel, with the COXIV-myc level of each uninduced cell line set to 100%. The level of COXIV-myc for each sample was normalised to its respective EF1a signal. Data are presented as mean values with error bars corresponding to the standard deviation of the mean from three independent biological replicates. Significance calculated using a two-tailed unpaired *t*-test, with the *p* values calculated indicated in the graph. **b** Immunoprecipitation (IP) using ubiquitin-binding Ubiqapture beads of whole cell extracts of the uninduced and induced (3d) ATOM69 RNAi cell line that expresses ΔMTS-COXIV-myc. 5% whole cell extracts (IN), 5% of the unbound protein fraction (UB) and 50% of the final eluate (IP) were separated by SDS PAGE. The resulting immunoblots were probed with an antiserum against ubiquitin (left panel, Ub) or for ΔMTS-COXIV-myc (right panel, myc). ATOM40 (bottom panel) is used as a negative control. Arrow indicates a band of monoubiquitinated ΔMTS-COXIV-myc detectable by both anti-Ub and anti-myc. This IP analysis was repeated independently twice with similar results. **c** IP using myc conjugated beads with whole cell extracts of the uninduced and induced (3d) ATOM69 RNAi cell line that expresses ΔMTS-COXIV-myc. 5% whole cell extracts (IN), 5% of the unbound protein fraction (UB) and 50% of the final eluate (IP) were separated by SDS-PAGE. The resulting immunoblots were probed for the myc-tagged COXIV variants (top panel), TbUbL1 (middle panel) and VDAC, which serves as a loading control (bottom panel). This IP analysis was repeated independently twice with similar results. Source data are provided as a Source Data file.

ΔMTS-COXIV-myc is also reduced albeit to a slightly lesser extent. This indicates that ΔMTS-COXIV-myc is still a recognised substrate of the degradation pathway triggered by ablation of ATOM69, showing that the presequence cannot be the main feature by which the degradation system recognises mislocalised mitochondrial proteins. Next, we performed a pull-down experiment with cells expressing ΔMTS-COXIV-myc in the presence and absence of ATOM69, using ubiquitin binding domain-conjugated beads. There is a general increase of ubiquitinated proteins in the whole cell lysate in the absence of ATOM69 compared to when ATOM69 is present, as evidenced by the more intense smear detected in this lane with an antibody directed against ubiquitin (Fig. 8b, left panel). Moreover, when the blot was analysed with anti-myc antibodies, we detect a band possibly representing monoubiquitinated ΔMTS-COXIV-myc as well as a smear for polyubiquitinated ΔMTS-COXIV-myc only in the absence of ATOM69 (Fig. 8b, right panel). Thus, ΔMTS-COXIV-myc is ubiquitinated upon triggering the degradation pathway by ablation of ATOM69. This suggests that ubiquitinated ΔMTS-COXIV-myc is an intermediate in the degradation pathway that removes mislocalised mitochondrial proteins. Finally, we performed a pull-down experiment with the same cell lines, this time using myc-conjugated beads. While myc-tagged COXIV was efficiently recovered in both eluates, a small amount of TbUbL1 was found in the pulled down fraction exclusively in the case when ATOM69 has been ablated (Fig. 8c). This indicates

that TbUbL1 indeed is able to interact with ΔMTS-COXIV-myc after its release from the nucleus. The low intensity of the TbUbL1 band can be explained by the fact that TbUbL1 is expected to only interact transiently with its substrates. It is however not possible from this experiment to discern whether the fraction of ΔMTS-COXIV-myc that binds to TbUbL1 is also ubiquitinylated. In summary, the results in Fig. 8 are consistent with the idea that substrates of the degradation pathway triggered by the ablation of ATOM69 are ubiquitinated, and that TbUbL1, after its translocation into the cytosol, binds these substrates and transfers them to the proteasome for degradation.

## Discussion

We have discovered a pathway in *T. brucei* that removes mistargeted aggregation prone mitochondrial proteins that accumulate in the cytosol upon an import defect. The pathway consists of at least three components, TbUbL1, TbE3HECT1 and Tb927.9.7200, excluding the proteasome (for an overview see Fig. 9). We believe this pathway is the first MQC system that has been characterised outside yeast and metazoans. Interestingly, the MQC pathway is specifically triggered after ablation of ATOM69, but not after ablation of other ATOM complex subunits, such as ATOM46 and ATOM12. This is surprising since ablation of ATOM69 does not cause a stronger import defect than is seen for the other subunits (Fig. 1). ATOM40 depletion also causes nuclear release of TbUbL1 to the cytosol (Supplementary Fig. 8b),

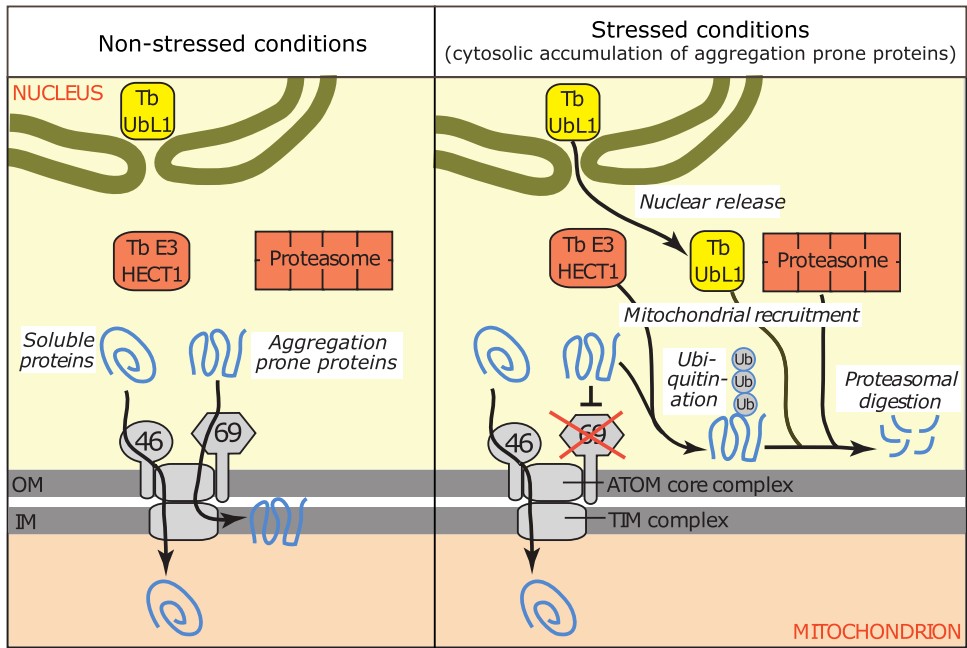

**Fig. 9 Schematic model depicting the MQC pathway triggered by ablation of ATOM69 in trypanosomes.** Ablation of atypical translocase of the outer membrane 69 (ATOM69) results in cytosolic accumulation of primarily hydrophobic proteins which triggers a quality control pathway that ultimately leads to the degradation of the mislocalised hydrophobic proteins. The pathway requires nuclear release of the ubiquitin-like protein 1, TbUbL1, the main subject of the present study (yellow). Moreover, TbUbL1, the ubiquitin ligase TbE3HECT1 (red), Tb927.9.7200 (not shown) and the proteasome (red) are recruited from the cytosol to the mitochondrion. The role of TbE3HECT1 may be to ubiquitinate the mislocalised hydrophobic proteins which subsequently may be recognised by TbUbL1 and transferred to the proteasome.

this however is likely an indirect effect since ATOM69 is also almost four-fold downregulated in an ATOM40 RNAi cell line (Fig. 1a). Interestingly, recruitment of MQC factors is not observed, possibly because ablation of ATOM40 interferes with import of all proteins, including the putative MQC factor Tb927.9.7200, and not just with a subset of substrates as is observed after ablation of ATOM69. This raises the question, what is so special about ATOM69? It has been shown that ATOM69 is an import receptor with a preference for hydrophobic proteins and thus a functional analogue of yeast Tom70[49,50]. Moreover, ATOM69 and Tom70 have the same molecular mass and contain a domain with multiple tetratricopeptide repeats (TPR). However, despite these shared features, the two proteins do not derive from the same ancestor but arose by convergent evolution. This is illustrated by their inverse topologies: ATOM69 is C-terminally anchored in the OM whereas Tom70 has its transmembrane domain at the N-terminus[50]. It has recently been shown that the essential in vivo role of yeast Tom70 is surprisingly not linked to its receptor function, but instead to protect the cytosol from proteotoxic stress caused by cytosolic accumulation of non-imported hydrophobic and other aggregation prone proteins[65]. More precisely, a proteophilic zone at the OM-cytosol boundary is protected from proteotoxic stress by Tom70-mediated recruitment of cytosolic chaperones to the mitochondrial surface. Interestingly, ATOM69 contains an N-terminal CS/Hsp20-like chaperone-binding domain, the removal of which only marginally slows down normal growth (Supplementary Fig. 9b), but leads to a growth arrest at elevated temperature[49]. Thus, as in the case of Tom70, ATOM69 is expected to recruit cytosolic chaperones to a putative proteophilic zone around the *T. brucei* mitochondrion. Loss of ATOM69 may prevent this recruitment and, as a consequence, cause proteotoxic stress in this zone. None of the other ATOM subunits, including the second trypanosomal protein import receptor ATOM46[49,50], contain recognisable chaperone

binding sites. Thus, ATOM69 depletion, in contrast to ablation of other ATOM subunits, may result in a depletion of cytosolic chaperones in the putative proteophilic zone around the OM, causing aggregation of mostly hydrophobic and other aggregation prone precursor proteins. This may in turn trigger the described ATOM69-depletion specific recruitment of MQC subunits to the mitochondrion (Fig. 2 and Supplementary Fig. 2). The connection of ATOM69 depletion to disturbance of a putative proteophilic zone around the OM is supported by the observation that lack of the N-terminal CS/Hsp20-like chaperone-binding domain alone is sufficient to trigger the release of TbUbL1 to the cytosol, and thus likely the whole pathway, even though it does not stop growth of the cells (Fig. 4b). Based on these results, it would be interesting to investigate whether tampering with Tom70 triggers a specific MQC pathway in the well-studied yeast system. Such an analysis could indicate whether a dedicated MQC pathway triggered by depletion of cytosolic chaperones from the proteophilic zone around the OM is a more general feature of eukaryotes.

When we compare the MQC pathways that result in the degradation of precursor proteins that accumulate in the cytosol in different eukaryotes, we find many common features. Thus, triggering of the pathway by impairment of mitochondrial protein import, recruitment of MQC factors to mitochondria and ubiquitin-mediated degradation of mislocalised mitochondrial proteins in the cytosol by the conserved proteasome are very similar in all systems. However, the individual components required for ubiquitination, as well as other essential factors, and the mechanisms by which they act, are not conserved.

Tb927.9.7200 is a non-essential kinetoplastid-specific protein and is the only protein of the three identified MQC components which has a mitochondrial localisation. Counterintuitively, the cellular level of this protein decreases upon import stress, whereas the presumably imported fraction present in the crude mitochondrial pellet increases (Fig. 2b). Like that seen for the other MQC components, ablation of Tb927.9.7200 inhibits the removal

of mistargeted mitochondrial proteins that accumulate in the cytosol. Presently, we do not understand the specific role it plays in the MQC pathway.

The kinetoplastid-specific TbE3HECT1 is recruited to the mitochondrion in the absence of the import receptor ATOM69, and likely adds ubiquitin to mislocalised mitochondrial proteins (Fig. 9). Due to its high molecular weight of 470 kDa, TbE3-HECT1 proved refractory to further experimental analysis.

The main focus of our work was on the kinetoplastid-specific TbUbL1 that, in response to ablation of ATOM69, was released from the nucleus and recruited to the mitochondrion (Fig. 9). Proteins with UBL domains are known to participate in MQC processes in other organisms. Examples include the conserved ubiquilins[24,25] and Ubx2[19]. In yeast and humans, ubiquilins direct mitochondrial membrane proteins for proteasomal degradation when they accumulate in the cytosol[24,25]. Thus, there is a functional connection between TbUbL1 and ubiquilin as both proteins appear to be implicated in MQC processes dealing predominantly with hydrophobic mitochondrial proteins. In yeast, the integral OM protein Ubx2 recruits the AAA-ATPase Cdc48 to the mitochondrial TOM complex to promote mitoTAD, and to the ER-associated degradation system (ERAD) to extract substrates that subsequently are targeted for degradation[19]. However, TbUbL1 is not a typical ubiquilin as it lacks a UBA domain typical of this protein family, and, unlike Ubx2, which has two transmembrane domains, it is a soluble protein.

TbUbL1 is recovered in a complex that includes ubiquitin under both normal and stress conditions (Fig. 5). Moreover, the protein contains a PIM motif, which suggests it may interact with the proteasome when ATOM69 is absent. From these results, we conclude that the role of TbUbL1 in the MQC pathway may be to recognise cytosolic substrates that require ubiquitination or that have been ubiquitinated, and to hand them over to the proteasome (Fig. 9). This is supported by the observation that ubiquitination of import-deficient COXIV, its interaction with TbUbL1 and finally its degradation, are stimulated by ablation of ATOM69. Possible candidates mediating ubiquitination of MQC substrates are either TbE3HECT1, which is essential for efficient degradation of the model substrate FtsH, or Tb927.10.780, a putative RING domain-containing E3 ubiquitin ligase that is constitutively associated with TbUbL1 (Fig. 5).

An early reaction to protein import stress is the release of at least a fraction of TbUbL1 from the nucleus to the cytosol. In line with its nuclear localisation under unstressed conditions, TbUbL1 contains a pat7-type monopartite NLS (Fig. 3c). Inhibition of nuclear release of TbUbL1 by LMB indicates its export is mediated by the CRM1 receptor for substrate nuclear export signals[60,61]. However, how the process is triggered and regulated is presently unknown. An attractive but purely speculative possibility for regulation would be that an as-yet-unidentified nuclear export signal in TbUbL1 is modulated by a reversible posttranslational modification such as phosphorylation. Our study shows that the TbUbL1 remains sequestered in the nucleus under non-stressed conditions. We cannot presently exclude that the protein has also a function in the nucleus, where it is associated with all five subunits of the conserved replication factor C involved in DNA replication[66]. However, individual RNAi-mediated ablation of TbUbL1 does not affect growth (Supplementary Fig. 5), suggesting it has neither an essential role in DNA replication nor cell cycle progression.

Nuclear release of TbUbL1 is required for the function of the ATOM69 depletion triggered MQC pathway in *T. brucei*. Relocalisation of proteins to and from the nucleus is also essential for MQC pathways that have been described in other organisms, for example *C. elegans* ATFS-1 relocates from the mitochondria to the nucleus upon mitochondrial dysfunction and acts as a transcription factor[28]. It activates the largely conserved UPR[mt] response that in humans and yeast is mediated by ATF4/ATF5 or HAP complex[26,31]. Moreover, impaired presequence processing in yeast induces a very early form of a UPR[mt] pathway, with the nuclear transcription factor Rox1 relocating to mitochondria, where it has a TFAM-like function in supporting mitochondrial transcription and translation[63]. In contrast to ATFS-1 and Rox1, which ultimately affect nuclear or mitochondrial transcription, the pathway mediated by TbUbL1 is purely posttranscriptional. The abundance of trypanosomal MQC factors does not change upon stress, rather a fraction of the protein partially relocates from the nucleus to mitochondria, in the case of TbUbL1, or from the cytosol to the proximity of the mitochondrion, in the case of TbE3HECT1 and the proteasome.

In the present work, we have studied a MQC pathway that is characterised by the nuclear release of TbUbL1 and that specifically responds to the absence of ATOM69, or more precisely to the absence of its N-terminal CS/Hsp20-like chaperone binding domain. These results suggest that the primary trigger of the pathway might be cytosolic aggregates of non-imported proteins. However, it is clear that other MQC pathways also exist in *T. brucei* which might be triggered by more general or different kinds of mitochondrial protein import stresses. In fact we have previously shown that depletion of ATOM40 results in efficient proteasomal degradation of precursor proteins that accumulate in the cytosol[52]. Thus, while nuclear release and mitochondrial recruitment of TbUbL1 appear to be closely linked to an absent or functionally impaired ATOM69, we cannot exclude that the other two factors that contribute to the pathway described in this study, mitochondrial Tb927.9.7200 and TbE3HECT1, may also be involved in other potentially overlapping MQC processes that all converge on the cytosolic proteasome.

Presently, we do not know what the physiological function of trypanosomal MQC pathways is. During its parasitic life cycle, *T. brucei* is challenged by many different environmental conditions. Unlike cells grown in vitro, the procyclic cells used in this study encounter significant temperature fluctuations in the tsetse fly, which cannot control its body temperature. This may temporarily impair mitochondrial protein import. The parasite also undergoes a programme of vast morphological and metabolic differentiation in response to environmental stimuli. In particular, during the latter life cycle stages in the tsetse fly, the citric acid cycle and oxidative phosphorylation are completely downregulated and the mitochondrion is reduced in volume[67]. Thus, it is possible that these differentiation steps result in a period of import stress where superfluous mitochondrial proteins accumulate in the cytosol and require specific degradation. MQC pathways connected to mitochondrial protein import stress may increase the fitness of the parasite population under such conditions. In fact it has been proposed that trypanosomal differentiation processes might have their origin in the repurposing of generic stress responses[34].

In the last few years, a large variety of MQC pathways have been investigated in both unicellular and multicellular eukaryotes. However, these studies were essentially restricted to yeast and mammals which belong to the same eukaryotic supergroup, the Opisthokonts, and therefore only cover a narrow branch of the eukaryotic diversity. Trypanosomes are a subgroup of the paraphyletic Discoba supergroup, which is highly diverged from Opisthokonts[33]. The fact that we find a MQC pathway in these cells that can counter the potentially toxic accumulation of mitochondrial proteins in the cytosol suggests that such pathways are required in all eukaryotes.

All MQC systems studied to date that act in the cytosol employ the ubiquitin system and the proteasome, both of which were already present in the last eukaryotic common ancestor (LECA), suggesting that at least a basic form of a MQC system was operational in LECA.

However, even though MQC systems have been studied only in a few Opisthokonts, and now one in *T. brucei*, we do see a great variety in the components making up these systems and in the mechanisms of how they operate. Thus, while many components of the MQC systems are based on the same toolbox inherited from LECA, they likely independently evolved in different phylogenetic groups.

## Methods

**Transgenic cell lines.** Transgenic *T. brucei* cell lines were generated using the procyclic strain 29–13[68]. Procyclic forms were cultivated at 27 °C in SDM-79[69] (Bioconcept) supplemented with 10% (v/v) fetal calf serum (FCS, Sigma Aldrich) containing G418 (15 µg/ml, Gibco), hygromycin (25 µg/ml, InvivoGen), puromycin (2 µg/ml, InvivoGen), blasticidin (10 µg/ml, InvivoGen) and phleomycin (2.5 µg/ml, LifeSpan BioSciences) as required. RNAi or protein expression was induced in cell lines by adding 1 µg/ml tetracycline to the medium.

Single and double RNAi cell lines were prepared using a pLew100-derived vector containing a drug resistance gene, with the generation of a stem-loop construct occurring by the insertion of the RNAi inserts in opposing directions. The loop is formed by a 460 bp spacer fragment. RNAi against the 3′ UTRs of ATOM69 (Tb927.11.11460) and ATOM14 (Tb927.11.7780), and the ORFs of ATOM12 (Tb927.8.4380) and ATOM40 (Tb927.9.9660) have been described previously[46,49,50]. RNAi plasmids were prepared targeting the indicated nucleotides of the ORFs of Tb927.11.4130 (nt 550–970), Tb927.8.1590 (nt 2878–3424), Tb927.9.7200 (nt 241–835), Tb927.10.2290 (nt 726–1228), Tb927.10.6300 (MRPS5, nt 436–890) and Tb927.9.13820 (KMP11, entire ORF). RNAi efficiency was verified by RNA extraction and Northern blot, as detailed in[70].

To produce the plasmids for the ectopic expression of C-terminal triple c-myc-tagged Tb927.11.4130 (TbUbL1), Tb927.9.7200 and Tb927.1.4100 (COXIV), the complete ORFs were amplified by PCR. The PCR product was cloned into a modified pLew100 vector[68,71] which contains a puromycin resistance gene as well as a triple epitope tag[72]. A mutant of Tb927.11.4130 containing a nuclear localisation sequence of the La protein (Tb927.10.2370)[64] at the C-terminus of the ORF, and a mutant of COXIV (Tb927.1.4100) lacking the previously defined MTS sequence at the N-terminus[62], were constructed by PCR, and cloned into the same expression vector. One FtsH allele was tagged in situ at the C-terminus with a triple HA-tag via a PCR approach[72], using pMOTag vector containing a phleomycin resistance cassette. Ectopic expression constructs for untagged FL-ATOM69 and ΔN103-ATOM69 were generated from ectopic expression c-myc tagging plasmids described previously[49]. Protein expression was verified by SDS-PAGE and immunoblotting of cell lysates.

**Phenotypic analysis.** Cells were induced and treated with 500 nM MG132 (Merck Millipore, 474790), 180 ng/ml LMB (Sigma Aldrich, L2913) or 10 µM CCCP (Thermo Fisher, L06932) as indicated. For immunoblots, $2 \times 10^6$ cells were analysed per sample. The following polyclonal antibodies were used: voltage dependent anion channel (VDAC) (1:1000)[73], ATOM40 (1:10,000)[50], ATOM69 (1:1000)[50], COXIV (1:1000)[74], cytochrome *c* (1:100)[75], TimRhomI (1:500)[76], mtHsp60 (1:500)[77], Alba3 (1:500)[78]. The polyclonal TbUbL1 rabbit antisera used for immunoblots was produced commercially by Eurogentec using peptide sequence CSEISGNHRSSEHNAG. The antibody was affinity purified against the antigen by Eurogentec, and used at a 1:50 dilution. Commercially available antibodies were used as follows: mouse c-Myc (Invitrogen, 132500; 1:2000), mouse HA (Enzo Life Sciences AG, ENZ-ABS118; 1:5000), rabbit anti-Ub (Proteintech, 10201-2-AP; 1:500) and mouse eukaryotic elongation factor 1a (EF1a) (Merck Millipore, 05–235; 1:10,000). Secondary antibodies for immunoblot analysis were IRDye 680LT goat anti-mouse, and IRDye 800CW goat anti-rabbit (both LI-COR Biosciences PN 926–68020, PN 926–32211; 1:20,000). Quantification of western blots was done using an Licor Odyssey Imaging System with software version 2.1.15 and Image Studio version 5.2.5. For immunofluorescence analysis, cells were fixed with 4% paraformaldehyde in PBS, permeabilised with 0.2% Triton-X100 in PBS and blocked with 2% BSA. Primary antibodies used were mouse anti-c-Myc (1:50) and rabbit anti-ATOM40 (1:1000), and secondary antibodies were goat anti-mouse Alexa Fluor 596 and goat anti-rabbit Alexa Fluor 488 (both ThermoFisher Scientific; 1:1000).

Slides were mounted with VectaShield containing 4′,6-diamidino-2-phenylindole (DAPI) (Vector Laboratories). Images were acquired with a DFC360 FX monochrome camera (Leica Microsystems) mounted on a DMI6000B microscope (Leica Microsystems). Images were analysed using LAS AF X product version 3.6.20104.0 software (Leica Microsystems) and ImageJ version 2.10./1.53c; Java 1.8.0_172 [64-bit] (NIH).

**Immunoprecipitations.** $1 \times 10^8$ cells expressing the tagged protein of interest were either solubilised for 15 min on ice in 20 mM Tris-HCl pH7.4, 0.1 mM EDTA, 100 mM NaCl, 25 mM KCl containing 1% (w/v) digitonin (Biosynth, 103203) and 1X Protease Inhibitor mix (Roche, EDTA-free, 11873580001), or by sonication on ice (3 x 10 s, with 30 s break) after lysis in 60 mM Tris-HCl, pH7.4, 6 mM EDTA, 200 mM NaCl, 1% Triton X100 and 1X Protease Inhibitor mix. After centrifugation (15 min, 20,000 g, 4 °C), the lysate (IN) was transferred to 50 µl of bead slurry, which had been previously equilibrated with respective lysis buffer. The bead slurries used were c-myc-conjugated (EZview red rabbit anti-c-myc affinity gel, Sigma Aldrich, E6654) or agarose goat anti-myc tag antibody, Abcam, ab1253) and ubiquitin binding domain (UBD)-conjugated matrix (Ubiqapture-Q, Enzo Life Sciences, BML-UW8995A-0001). After incubating at 4 °C for 2 h, the supernatant containing the unbound proteins (UB) was removed, the bead slurry was washed three times with lysis buffer and the bound proteins were eluted by boiling the resin for 10 min in 2% SDS in 60 mM Tris-HCl pH 6.8 (IP). 5% of both the input and the flow through samples were retained and subjected to SDS-PAGE and Western blotting, along with 50% of the IP sample. For IPs with cells expressing ΔMTS-COXIV-myc, cells were induced for 3 days in total, with 500 nM MG132 added to the cells on d2 post induction.

**SILAC proteomics.** Cells were washed in PBS and taken up in SDM-80[79] supplemented with 5.55 mM glucose, either light ($^{12}C_6/^{14}N_X$) or heavy ($^{13}C_6/^{15}N_X$) isotopes of arginine (1.1 mM) and lysine (0.4 mM) (Euroisotop, CNLM-291-H-0.5, CNLM-539-H-1) and 10% dialysed FCS (BioConcept). To guarantee complete labelling of all proteins with heavy amino acids, the cells were cultured in SILAC medium for 6–10 doubling times.

RNAi cell lines targeting the ORFs of ATOM12 (Tb927.8.4380)[50], ATOM14 (Tb927.11.5600)[50] and ATOM19 (Tb927.9.10560)[51], the ATOM69 3′ UTR[49], and a conditional knockout ATOM11 (Tb927.10.11030)[50] cell line all generated previously were used to generate SILAC depletomics samples. For their induction time with tetracycline, see legends. $1 \times 10^8$ uninduced and $1 \times 10^8$ induced cells were harvested and mixed. Crude mitochondria-enriched pellets were obtained by incubating $2 \times 10^8$ cells on ice for 10 min in 0.6 M sorbitol, 20 mM Tris-HCl pH 7.5, 2 mM EDTA pH 8 containing 0.015% (w/v) digitonin, and centrifugation (5 min/6800 g/4 °C). Either the mixed whole cells, or digitonin-extracted mitochondria-enriched pellets generated from these mixed cells, were then analysed.

For SILAC-IP experiments, cells were induced for 4 days. $1 \times 10^8$ uninduced and $1 \times 10^8$ induced cells were harvested, mixed and subjected to CoIP as described above.

All SILAC experiments were performed in three biological replicates including a label-switch, and analysed by liquid chromatography-mass spectrometry (LC-MS).

**LC-MS and data analysis.** LC-MS analyses of samples derived from SILAC RNAi experiments, including sample preparation and mass spectrometric measurements, were performed as described previously[49,52]. Mitochondria-enriched fractions were analysed following either a gel-based (ATOM11, ATOM12, and ATOM14 RNAi experiments[52] or a gel-free approach (ATOM19 RNAi and ATOM69 RNAi+/− LMB experiments[49]). Proteins of whole cell extracts prepared from mixed uninduced and induced ATOM69 RNAi cells were digested using trypsin, followed by fractionation of the resulting peptide mixtures by high-pH reversed-phase liquid chromatography prior to LC-MS analysis as reported[52]. All samples of ATOM RNAi experiments were analysed on an Orbitrap Elite instrument (Thermo Fisher Scientific, Germany). Eluates of TbUbL1-myc SILAC-IP experiments were analysed on a Q Exactive Plus (Thermo Fisher Scientific, Germany). LC-MS sample preparation and analysis were performed as described before[70].

The proteomics software package MaxQuant with its integrated search engine Andromeda was used for protein identification and SILAC-based relative quantification (versions 2.0.2.0 for ATOM69 RNAi+/− LMB data, 1.6.10.43 for all other RNAi data, and 1.5.5.1 for IP data[80,81]). Mass spectrometric raw data were searched against a fasta file containing all protein sequences for *T. brucei* TREU927 as provided by the TriTrypDB (version 8.1; https://tritrypdb.org/). MS data acquired in this work for mitochondria-enriched fractions from ATOM11, ATOM12, ATOM14 and ATOM19 RNAi as well as whole cell extracts from ATOM69 RNAi experiments were processed together with datasets previously published for mitochondrial fractions of ATOM40[52], ATOM46 and ATOM69[49] RNAi experiments. Protein identification and quantification was based on ≥ 1 unique peptide and ≥ 1 ratio count, respectively. For all other parameters, MaxQuant default settings were used, including carbamidomethylation of cysteine as fixed modification, N-terminal acetylation and oxidation of methionine as variable modifications, Lys4/Arg6 as medium-heavy and Lys8/Arg10 as heavy labels. To identify proteins with significantly altered abundance upon RNAi-induced ablation of the different ATOM components or following LMB treatment of induced ATOM69 RNAi cells, we followed the 'linear models for microarray data' (limma) approach (two-sided test), a moderated t-test that adjusts a protein's variance in ratios between replicates towards the average ratio variance of the entire dataset[53,54]. *P*-values were determined for proteins quantified in ≥ 2 replicates (ATOM69 RNAi+/− LMB data: 3/3 replicates) and adjusted for multiple testing according to Benjamini-Hochberg[82]. The rank sum method (one-sided test)[83], as implemented in the R package 'RankProd' (version 3.11)[84], was applied to calculate *p*-values for the enrichment of proteins that were quantified in 3/3 replicates of TbUbL1-myc SILAC-IP experiments. The rank sum, defined as the arithmetic mean of the ranks of a protein in all replicates, was converted into a *p*-value and a false discovery rate. Proteins with the smallest rank sum are the most likely to be significantly enriched. MaxQuant result files were parsed with python using pandas (version 1.2.1; https://pandas.pydata.org/) as well as numpy (v. 1.19.2; https://numpy.org/), seaborn (v. 0.11.1; https://seaborn.pydata.org/), scipy (v. 1.5.2; https://www.scipy.org/), scikit-learn (v. 0.24.1; https://scikit-learn.org/stable/) and

matplotlib (v. 3.3.2; https://matplotlib.org/) for data analysis and visualisation. Results of the quantitative LC-MS analyses of ATOM RNAi, TbUbL1-myc IP and ATOM69 RNAi+/− LMB experiments are provided in Supplementary Data 1–3. To analyse if depletion of individual ATOM subunits has different effects on the mitochondrial proteome, mitochondrial proteins significantly altered in abundance (i.e., $p$-value < 0.01) at least once upon depletion of either ATOM19, ATOM12, ATOM11, ATOM46 or ATOM69 were subjected to agglomerative hierarchical clustering following the Ward's method, based on a Spearman correlation-based distance matrix. The Spearman rank correlation was computed using the function "DataFrame.corr" of the python pandas package (v. 1.2.1). ATOM40 and ATOM14 RNAi data were omitted from this analysis since depletion of these subunits resulted in severe effects on the mitochondrial proteome (see Fig. 1b), hampering the identification of more subtle differences between the remaining five ATOM subunits. The number of clusters selected for this dataset (i.e., $k = 5$) was based on the Davies-Bouldin index. For visualisation, a row-wise z-score transformation of mean $log_2$ normalised protein abundance ratios was performed. Information about mitochondrial proteins present in the clusters are provided in Supplementary Data 1b.

**Miscellaneous**. Uncropped and unprocessed scans of all blots are provided the Source Data file or as a supplementary fig. in the Supplementary Information.

**Reporting summary**. Further information on research design is available in the Nature Research Reporting Summary linked to this article.

## Data availability

The mass spectrometric data have been deposited to the ProteomeXchange Consortium[85] via the PRIDE[86] partner repository and are available with the identifier PXD027739 (ATOM subunits RNAi data; Supplementary Data 1a), PXD027652 (TbUbL1-myc IP data; Supplementary Data 2) and PXD031888 (LMB data; Supplementary Data 3). TriTrypDB v. 8.1. was used (https://tritrypdb.org/). Source data are provided with this paper.

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

## Acknowledgements

We thank Bettina Knapp and Kurt Lobenwein for support by LC-MS analyses, Christian D. Peikert for initial MS data analyses and the PRIDE team for data deposition to the ProteomeXchange Consortium, and Isabel Roditi for the gift of the Alba3 antibody. Work in the lab of AS was supported by grant 175563 and in part by the NCCR "RNA & Disease" both funded by the Swiss National Science Foundation. Work in the lab of BW was supported by the Deutsche Forschungsgemeinschaft (DFG, German Research Foundation) Project ID 403222702/SFB 1381 and the Germany's Excellence Strategy (CIBSS – EXC-2189 – Project ID 390939984).

## Author contributions

C.E.D. and S.O. conceptualized, planned, executed, and analyzed most experiments, C.E.D. with a focus on biochemical and cell biological experiments and S.O. with a focus on proteomics. C.E.D., A.S., and B.W. wrote the original draft, reviewed and edited all versions of the manuscript. J.M. planned and executed experiments. T.O. quantified the immunofluorescence results. C.v.K. provided growth curves and analyzed immuno-fluorescence results. W.W.D.M. and J.Z. performed MS data analysis and statistics. A.S. and B.W. acquired funding for the project.

## Competing interests

The authors declare no competing interests.
