## [Peer Review File · Nature Communications]

Peer review comments, first round review –

Reviewer #1 (Remarks to the Author):

Reviewer's comment

This is a very interesting paper consisting of several novel findings. The paper describes that depletion of ATOM69, a receptor translocase of the ATOM complex in *Trypanosoma brucei*, triggers a protein quality control mechanism to degrade unimported mitochondrial proteins. Using multiple SILAC proteomics, authors identified a group of mitochondria-associated proteins with increased relative abundance due to ATOM69 depletion. Among these, TbUbl1, a ubiquitin-like protein that is released from the nucleus during Atom69 RNAi and is required for degradation of unimported precursor proteins, like cytochrome oxidase subunit IV (COXIV) and FtsH. TbUbl1 is associated with a E3 ubiquitin ligase domain containing protein, ubiquitin, and some other novel proteins in *T. brucei*. These results showed that like in yeast and metazoans, mitochondrial translocation-associated degradation (mTAD) pathway exists in an early divergent parasitic protozoan such as *T. brucei*. Most of the work are wisely performed. There are few concerns regarding the interpretation and conclusion of some the experiments and these are listed below.

1. The title of the paper is "Mistargeting of hydrophobic mitochondrial protein—", however authors validated their SILAC data using COXIV that does not have any transmembrane domain but possesses presequence. In this manuscript (Lines 103-107) and in their previous publication authors classified that ATOM46 has substrate specificity for presequence-containing hydrophilic proteins, whereas ATOM69 has substrate specificity for presequence-lacking hydrophobic proteins. However, it has been considered that COXIV is the substrate for Atom69 receptor. This is somewhat confusing unless it must be assumed that there is a huge overlap among ATOM46 and ATOM69 substrates. In this respect authors could modify their statement regarding substrate-specificity and functional similarity of ATOM69 with fungal Tom70.

A large number of essential mitochondrial matrix proteins are hydrophilic and have presequence. If these are the likely substrate for ATOM46, it is expected that depletion of ATOM46 would be more growth inhibitory than the other receptor. However, the opposite results have been shown. Therefore, it further indicates a larger overlap of the substrate proteins for these receptors.

2. Fig. 4A, TbUbl1-myc staining appears more intense in the nucleus in ATOM69 RNAi cells. Since the levels of TbUbl1 in total cell lysate was unchanged, and TbUbl1 is released in the cytosol due to ATOM69 RNAi, it is expected that the nuclear staining should be reduced. Quantitation of nuclear and cytosolic intensities from multiple cells are necessary. Immunoblot analysis of the nuclear and cytosolic fractions could also be supportive.

3. Fig. 5B, were the intensities of FtsH bands normalized with the corresponding EF1a band intensities? Why the levels of ATOM69 increased after LMB treatment? There are multiple FtsH isomers in *T. brucei*. Which of these was expressed? Does overexpression of this mitochondrial protease affect cell growth?

4. Fig. 6B, TbUbl1-NLS-myc intensity in the nucleus appeared relatively less in the presence of ATOM69 RNAi. Does it mean that despite an additional NLS some cytosolic release occurred? The distribution ration in cytosol vs nucleus should be quantitated from multiple cell images. What is the distribution ratio of TbUbl1 with one and two NLS in cytosol/nucleus?

5. Fig. 7, Why COXIV was used instead of FtsH for ubiquitination studies? It has been shown previously that COXIV precursor protein accumulates due to ATOM69 RNAi, however that has not been seen in Fig. 7A. The IP lanes in the left panel of Fig 7B are practically similar in the presence and absence of ATOM69 RNAi. How it looks when the full blot was Probed with anti-myc? What percentage of COXIV was ubiquitinated? Chemical cross-linking could increase the level of TbUbl1 in the IP in Fig. 7C.

Minor Points

- ATOM69 depletion showed relatively lower effect on depletion of mitochondrial proteins in comparison to ATOM40 and ATOM14 RNAi but increased more non-mitochondrial proteins in the crude mitochondrial fraction. From this observation authors conclude that depletion of ATOM69 triggers a specific stress response (Line 150-151). At this stage of work this statement is an over interpretation.
- Tb927.9.7200 has MTS, why it is excluded from the mito reference protein?
- Line 129, a linear model

Reviewer #2 (Remarks to the Authors)

Mitochondria import most of their proteins post-translationally. Defects in the import machinery can lead to the accumulation of mitochondrial precursor proteins in the cytosol. To avoid toxic effects of these precursors, cells recognize precursors and rapidly degrade them. Despite its importance for cellular functionality, the molecular biology of this degradation process is not known. In particular, it remains unclear how cells can distinguish toxic non-productive precursors that have to be degraded from productive precursors that are on transit to mitochondria. The study by Dewar and colleagues shows a beautiful mechanism which results in the specific depletion of cytosolic precursors from cells of the model organism *Trypanosoma brucei*. This parasite employs co-factors of the ubiquitin-proteasome system, in particular a protein which the authors called TbUbl1, that resides in the nucleus under non-stress conditions. Upon accumulation of mitochondrial precursors, TbUbl1 is released from the nucleus and induces the ubiquitination and degradation of precursor proteins. By use of very comprehensive proteome analyses in cells in which import components were depleted, the authors identified a large set of proteins which are specifically recruited onto the mitochondrial surface. This study is very inspiring and exciting. The close cooperation of the cytosol and the nucleus in the mitochondrial quality control was recently reported in yeast (Shakya, 2021. *Elife* 10:e61230). However, this study here goes far beyond any published articles as for the first time, it reports about a mechanism by which mitochondrial precursors can be specifically cleansed from stressed cells. Two major aspects should be addressed in order to even improve this exciting study:

1. One major observation of this study is the recruitment of PQC factors to the mitochondrial surface upon knock down of ATOM subunits. How these factors are directed to the outer membrane is unclear. It is possible that this recruitment is mediated simply by the binding of these factors to the translocation intermediates that are stalled on the outer membrane. Alternatively, they directly bind the outer membrane as a result of a stress response. In order to distinguish this, the authors might block protein synthesis prior to the experiment, e.g. for two hours before mitochondria of ATOM69-depleted cells are isolated. Under these conditions, translocation intermediates should be largely absent and thus, direct binding and precursor-mediated recruitment could be distinguished. Such an experiment should be analyzed by MS rather than by Western blotting in order to get a general overview.
2. The effect of the nuclear export inhibitor LMB as shown in Fig. 5 is impressive. Unfortunately, the authors only used this nice drug for a western blot with FtsH-HA. The study would be even more interesting, if the authors would perform MS of mitochondrial fractions derived from these LMB-treated cells. Thereby the spectrum of proteins that are degraded by the TbUbl1-dependent mechanism could be identified. This would be a nice, though not essential experiment. It also might help to exclude that the suppression of the FtsH-HA degradation observed is just the consequence of the inefficient knock-down of ATOM69 in the presence of LMB.

Minor points:

3. The nomenclature of proteins in Tb is really annoying as it is very difficult to remember these numbers. For proteins studied here, I recommend to come up with an acronym that can be used by others. Something like MQC and the molecular mass of the protein. Also the abbreviation of

UbL1 is not perfect as it might be confused with a ubiquitin which has a rather similar function in human cells.

4. The depletion of mitochondrial proteins as shown in the data of Fig. 1BC is impressive and very convincing. Since it is very difficult to have a closer look at the non-ATOM proteins for scientists who are not familiar with the trypanosome protein nomenclature, it would be interesting to add a figure to the supplement in which the authors analyzed, whether the depletion pattern from these different siRNAs are similar or all different, for example by a heat map of spearman correlations. Moreover it would interesting to know which class of proteins were predominantly depleted? The data set would allow conclusion on whether outer membrane proteins are more depleted than matrix proteins, or whether membrane proteins are more affected than hydrophilic proteins etc. This would be an information of general importance for a broad readership.

5. Fig. 7B shows a prominent modification of DMTS-COXIV-myc from which the authors suggest it is monoubiquitination. It also could be SUMO, or? See n. The prominent monoubiquitinated DMTS-COXIV-myc. Couldn't this be sumo (see Paasch et al., JBC 293, 599f)? Since it is unclear which modification it really is, the authors should better not label it as monoUb, or prove its ubiquitination.

6. In the abstract, the authors should better change the last sentence ,such pathways are an obligate feature of all mitochondria' for ,such pathways are an obligate feature of all eukaryotes' as the components described here act in the cytosol/nucleus.

7. Line 158: 'these proteins are degraded by the proteasome when their import is inhibited'. It is also possible that lower amounts of these proteins were synthesized. Such a repression of mitochondrial proteins was also observed in other systems, such as in yeast (e.g. Boos et al, Nature Cell Biology).

Reviewer #3 (Remarks to the Author):

Review of NCOMMS-21-31553

Summary: In this report by Dewar et al., the authors investigate the effects of knocking down the ATOM69 subunit of the mitochondrial protein import receptor complex in *T. brucei*. Depletion of ATOM69 uncovered a protein quality control pathway that likely functions in normal cells and led to the identification of three putative components of this pathway. While mitochondrial protein quality control mechanisms have been uncovered in other model organisms including yeast, nematodes and mammals, finding similar pathways in the highly diverged trypanosome suggest that mitochondrial quality control is a central feature of all eukaryotes. This work is significant and of interest to a wide range of cell biologists, warranting publication in Nature Communications.

Specific comments:

1. The authors conclude that ablation of ATOM69 triggers a specific stress response not seen when the other six subunits of the ATOM complex (atypical translocase of the outer membrane) are knocked down. This result is puzzling since loss of many of the other ATOM subunits also show reduced or absent ATOM69 levels, one might expect that a similar stress pathway is likewise activated. While their proteomic analyses clearly shows that loss of the other ATOM subunits does not lead to mitochondrial accumulation of TbUbl1, TbE3HECT1, or Tb927.9.7200, it does not rule out that these proteins are also required for the degradation of non-imported mitochondrial proteins when the other ATOM subunits are depleted. The authors do address this, albeit very briefly, in the Discussion. Perhaps determining directly whether TbFtsH-myc turnover requires TbUbl1, TbE3HECT1, or Tb927.9.7200 in ATOM11, 12, 40 or 46 knockdowns is warranted.

2. Along similar lines, the authors propose that ATOM69, like the yeast Tom70 counterpart, is required for the import of a subset of mitochondrial proteins—such as hydrophobic or aggregation-prone substrates. However, to probe the function of ATOM69 and TbUbl1 they use the small, hydrophilic substrate, COXIV, which based on their model is not imported via the ATOM69 pathway. While their results indeed show that loss of TbUBL1 leads to stabilization (and ubiquitination) of COXIV, it seems to argue against their idea of an ATOM69-specific pathway.

3. To show that nuclear release of TbUbl1 is required for mitochondrial protein turnover, the authors add a second NLS to TbUbl1 and overexpress this construct in cells. Consistent with their experiments inhibiting export (with LMB), the TbUbl1-NLS construct inhibits turnover. The authors conclude that they have constructed a dominant-negative version of TbUbl1, but this is not the only explanation for what is going on. My least favorite part of the manuscript overall is that the authors seem bound and determined to fit their results into the mitochondrial protein quality control 'box', instead of letting the results lead where they may. Nonetheless, I am confident that they are most likely correct in their interpretations and that this work, with a bit of mostly editing work, is suitable for publication.

4. Picky point: The authors use the term down regulated in several places in the manuscript. Since this term is likely to be misinterpreted by many to indicate transcriptional regulation, it is probably best to avoid.

We would like to thank the three reviewers for their fair and constructive comments. We are confident that we could respond to all of their criticisms and believe that the revised manuscript has been greatly improved. - Below please find our point by point responses.

REVIEWER COMMENTS

REVIEWER #1 (REMARKS TO THE AUTHOR):

THIS IS A VERY INTERESTING PAPER CONSISTING OF SEVERAL NOVEL FINDINGS. THE PAPER DESCRIBES THAT DEPLETION OF ATOM69, A RECEPTOR TRANSLOCASE OF THE ATOM COMPLEX IN TRYPANOSOMA BRUCEI, TRIGGERS A PROTEIN QUALITY CONTROL MECHANISM TO DEGRADE UNIMPORTED MITOCHONDRIAL PROTEINS. USING MULTIPLE SILAC PROTEOMICS, AUTHORS IDENTIFIED A GROUP OF MITOCHONDRIA- ASSOCIATED PROTEINS WITH INCREASED RELATIVE ABUNDANCE DUE TO ATOM69 DEPLETION. AMONG THESE, TBUBL1, A UBIQUITIN-LIKE PROTEIN THAT IS RELEASED FROM THE NUCLEUS DURING ATOM69 RNAI AND IS REQUIRED FOR DEGRADATION OF UNIMPORTED PRECURSOR PROTEINS, LIKE CYTOCHROME OXIDASE SUBUNIT IV (COXIV) AND FTSH. TBUBL1 IS ASSOCIATED WITH A E3 UBIQUITIN LIGASE DOMAIN CONTAINING PROTEIN, UBIQUITIN, AND SOME OTHER NOVEL PROTEINS IN T. BRUCEI. THESE RESULTS SHOWED THAT LIKE IN YEAST AND METAZOANS, MITOCHONDRIAL TRANSLOCATION-ASSOCIATED DEGRADATION (MTAD) PATHWAY EXISTS IN AN EARLY DIVERGENT PARASITIC PROTOZOAN SUCH AS T. BRUCEI. MOST OF THE WORK ARE WISELY PERFORMED. THERE ARE FEW CONCERNS REGARDING THE INTERPRETATION AND CONCLUSION OF SOME THE EXPERIMENTS AND THESE ARE LISTED BELOW.

1. THE TITLE OF THE PAPER IS "MISTARGETING OF HYDROPHOBIC MITOCHONDRIAL PROTEIN—", HOWEVER AUTHORS VALIDATED THEIR SILAC DATA USING COXIV THAT DOES NOT HAVE ANY TRANSMEMBRANE DOMAIN BUT POSSESSES PRESEQUENCE. IN THIS MANUSCRIPT (LINES 103-107) AND IN THEIR PREVIOUS PUBLICATION AUTHORS CLASSIFIED THAT ATOM46 HAS SUBSTRATE SPECIFICITY FOR PRESEQUENCE-CONTAINING HYDROPHILIC PROTEINS, WHEREAS ATOM69 HAS SUBSTRATE SPECIFICITY FOR PRESEQUENCE-LACKING HYDROPHOBIC PROTEINS. HOWEVER, IT HAS BEEN CONSIDERED THAT COXIV IS THE SUBSTRATE FOR ATOM69 RECEPTOR. THIS IS SOMEWHAT CONFUSING UNLESS IT MUST BE ASSUMED THAT THERE IS A HUGE OVERLAP AMONG ATOM46 AND ATOM69 SUBSTRATES. IN THIS RESPECT AUTHORS COULD MODIFY THEIR STATEMENT REGARDING SUBSTRATE-SPECIFICITY AND FUNCTIONAL SIMILARITY OF ATOM69 WITH FUNGAL TOM70.

We do not claim that ATOM69 or ATOM46 have exclusive substrate specificities, rather we always talk about substrate preferences. The reason is that in vivo import of most proteins depends to various extents on both receptors ¹. Thus, there is indeed a large overlap. A recent study in yeast suggests that the situation is similar for yeast Tom70 which, in vitro, has a preference for hydrophobic substrates, but in vivo is also required to import some presequence-containing proteins ². Regarding COXIV we have quantitative information, its import depends to equal parts on ATOM69 (50.3%) and ATOM46 (49.7%) ¹.

The reviewer is correct that COXIV is a soluble protein lacking transmembrane domains in *T. brucei*. However, when we analyzed its properties experimentally, we found that it does not behave like a typical soluble protein. In a carbonate extraction at high pH, the majority of COXIV fractionates with the pellet, even though soluble and peripheral membrane proteins are expected to be completely soluble under these conditions. Moreover, in a digitonin-based aggregation assay as described in ³, more than 30% of both native as well as tagged COXIV aggregate in 1% digitonin, whereas much less aggregation (below 13%) was seen for two integral membrane proteins (VDAC and TimRhom I) or two soluble proteins (mtHsp60 and Alba3). Thus, while COXIV lacks transmembrane domains, it is more aggregation prone than other soluble and even integral membrane proteins tested. This may explain why it requires both receptors for import.

We have added the results of the experiments described above as new supplementary Fig.11 to the revised manuscript (described in lines 343-354). Moreover, we have modified Fig. 6bc and Fig. 7bc and added the data for COXIV so that it can be directly compared to our other substrate FtsH.

Since most membrane proteins, the preferred substrates for ATOM69, are aggregation prone and the same seems to be the case for COXIV we have changed the title of our manuscript and replaced the term “hydrophobic with “aggregation prone”.

A LARGE NUMBER OF ESSENTIAL MITOCHONDRIAL MATRIX PROTEINS ARE HYDROPHILIC AND HAVE PRESEQUENCE. IF THESE ARE THE LIKELY SUBSTRATE FOR ATOM46, IT IS EXPECTED THAT DEPLETION OF ATOM46 WOULD BE MORE GROWTH INHIBITORY THAN THE OTHER RECEPTOR. HOWEVER, THE OPPOSITE RESULTS HAVE BEEN SHOWN. THEREFORE, IT FURTHER INDICATES A LARGER OVERLAP OF THE SUBSTRATE PROTEINS FOR THESE RECEPTORS.

As mentioned above there is indeed a large overlap. Many presequence-containing proteins while preferring the ATOM46 receptor also require ATOM69 (see ¹ for a quantitative analysis). A possible explanation could be the following: ATOM69, in contrast to ATOM46, has an N-terminal CS/Hsp20-like chaperone-binding domain which can recruit chaperones to the outer membrane. It is therefore possible that ATOM46-dependent substrates may profit from this chaperone recruitment even though they may not use ATOM69 as a receptor.

2. FIG. 4A, TBUBL1-MYC STAINING APPEARS MORE INTENSE IN THE NUCLEUS IN ATOM69 RNAI CELLS. SINCE THE LEVELS OF TBUBL1 IN TOTAL CELL LYSATE WAS UNCHANGED, AND TBUBL1 IS RELEASED IN THE CYTOSOL DUE TO ATOM69 RNAI, IT IS EXPECTED THAT THE NUCLEAR STAINING SHOULD BE REDUCED. QUANTITATION OF NUCLEAR AND CYTOSOLIC INTENSITIES FROM MULTIPLE CELLS ARE NECESSARY.

We agree and now provide a quantitative analysis of the immunofluorescence (IF) data shown in Fig. 4a. In the left panel of supplementary Fig. 7a we quantified the nuclear TbUbl1-HA signals in uninduced and induced ATOM69 RNAi cells. These signals were not significantly different. The picture presented in the original Fig. 4a was therefore not typical and we replaced it with a more typical one in the revised Fig 4a. Moreover we also quantified the ratio of nuclear to cytosolic signal of TbUbl1 in the two populations. This ratio was significantly different as would be expected due to the release of TbUbl1 in induced cells (supplementary Fig. 7a, right panel).

We would like to emphasize that it is not clear whether the partial release of TbUbl1 from the nucleus will indeed result in a reduced nuclear staining compared to unstressed cells. It needs to be considered that TbUbl1 has different binding partners under normal and stress conditions (see Fig. 5). It is therefore possible that the myc-tag is differentially masked in either of the two compartments or under normal or stress conditions, respectively.

Finally, even though not demanded by the reviewers, we also counted the fraction of cells that showed nuclear release of TbUbl1 to the cytosol in uninduced and induced ATOM69 RNAi cells. Practically this was done as follows: the release was scored from multiple IF images which were blinded, meaning that the person that analysed the images did not know which images represent the experiment and which ones the control. The results of the same type of analyses were also added to the IF pictures shown in Fig. 4, 6 and 7.

IMMUNOBLOT ANALYSIS OF THE NUCLEAR AND CYTOSOLIC FRACTIONS COULD ALSO BE SUPPORTIVE.

We extensively tried this. However, biochemical fractionation of nuclear and cytosolic fraction did not work in our hands. The problem was that soluble nuclear proteins which we used as markers were not retained in the nuclear fraction.

3. FIG. 5B, WERE THE INTENSITIES OF FTSH BANDS NORMALIZED WITH THE CORRESPONDING EF1A BAND INTENSITIES?

Yes, the FtsH-HA intensities were always normalized to EF1a intensities. This information has now been added to the corresponding figure legend.

WHY THE LEVELS OF ATOM69 INCREASED AFTER LMB TREATMENT?

We do not know why the levels of ATOM69 in the induced RNAi cell line are increased after LMB treatment. One possibility is that efficient RNAi may require export of factors from the nucleus.

THERE ARE MULTIPLE FTSH ISOMERS IN T. BRUCEI. WHICH OF THESE WAS EXPRESSED?

The accession number of the FtsH used in our study is Tb927.11.14730, which was termed FtsH14 in a previous publication⁴. We have added the corresponding reference to the revised manuscript.

DOES OVEREXPRESSION OF THIS MITOCHONDRIAL PROTEASE AFFECT CELL GROWTH?

No, it does not, there is no significant difference in growth between the FtsH-HA expressing cells and the parent cell line it is derived from (see growth curve Figure 1 below). FtsH-HA is an in situ tagged protein, thereby likely not overexpressed. We think adding this data to the manuscript would be tangential and therefore would like to provide it for the reviewer only.

Figure 1. Growth curves of uninduced ATOM69-RNAi cells and uninduced ATOM69-RNAi cells containing FtsH in situ 3x HA tagged at its C-terminus as indicated. Error bars (too small to be visible) correspond to the standard deviation ($n = 3$).

4. FIG. 6B, TBUBL1-NLS-MYC INTENSITY IN THE NUCLEUS APPEARED RELATIVELY LESS IN THE PRESENCE OF ATOM69 RNAI. DOES IT MEAN THAT DESPITE AN ADDITIONAL NLS SOME CYTOSOLIC RELEASE OCCURRED? THE DISTRIBUTION RATION IN CYTOSOL VS NUCLEUS SHOULD BE QUANTITATED FROM MULTIPLE CELL IMAGES. WHAT IS THE DISTRIBUTION RATIO OF TBUBL1 WITH ONE AND TWO NLS IN CYTOSOL/NUCLEUS?

We now provide a quantitative analysis of the immunofluorescence (IF) data shown in the two panels of Fig. 6a (new supplementary Figure 7b). However, it proved essentially impossible to measure a reliable cytosolic signal because it was too close to background staining suggesting that essentially no NLS-containing TbUbl1 is released in either of the two cell lines. Thus, we could not determine the ratio of nuclear to cytosolic signal of TbUbl1. Instead we quantified the intensity of nuclear TbUbl1 signal which was essentially identical in both cell lines.

5. FIG. 7, WHY COXIV WAS USED INSTEAD OF FTSH FOR UBIQUITINATION STUDIES?

As described above import of COXIV depends to equal parts on both ATOM69 and ATOM46. In line with this observation new supplementary Fig. 11 shows that COXIV does not behave like a typical soluble protein but is much more aggregation prone than typical soluble and even some membrane proteins (see above).

Binding of substrates to TbUbl1 can only be detected when the substrates accumulate in the cytosol. We chose COXIV for the experiment in the former Fig. 7 (Fig. 8 in the revised manuscript) because it has a previously mapped N-terminal mitochondrial targeting sequence. Removing this sequence allows for complete accumulation of the protein in the cytosol. We cannot do the same experiment with FtsH because it lacks a N-terminal targeting sequence and its targeting signal is unknown.

We nevertheless tried to do an analogous experiment with the FtsH substrate. Instead of using the in situ tagged version of FtsH, we overexpressed FtsH-myc under tet-control. The experiment was also done in the background of a cell line overexpressing HA-tagged ubiquitin, which should greatly facilitate detection of ubiquitinated proteins. However, neither experiment worked as upon expression of the substrate for three days (the time point for the experiment), it was found to be extensively digested.

IT HAS BEEN SHOWN PREVIOUSLY THAT COXIV PRECURSOR PROTEIN ACCUMULATES DUE TO ATOM69 RNAI, HOWEVER THAT HAS NOT BEEN SEEN IN FIG. 7A.

In the experiments shown in the former Fig. 7a ATOM69 (Fig 8a in the revised manuscript) RNAi was induced for three days which is too short to detect precursor accumulation. As previously published it takes 5-6 days of RNAi induction to see the COXIV precursor in ATOM69 RNAi cell line (see Fig. 3c in reference ⁵).

THE IP LANES IN THE LEFT PANEL OF FIG 7B ARE PRACTICALLY SIMILAR IN THE PRESENCE AND ABSENCE OF ATOM69 RNAI. HOW IT LOOKS WHEN THE FULL BLOT WAS PROBES WITH ANTI-MYC?

The full blot is shown below in Fig. 2. It will be added to the full scan of all gels that will be compiled to a supplementary Figure in case the paper will be accepted.

Figure 2. Full immunoblot probed for myc of the right panel of previous Fig. 7b (Fig. 8b in the revised manuscript).

WHAT PERCENTAGE OF COXIV WAS UBIQUITINATED?

It is very difficult to quantify these signals. The signal to noise ratio in our opinion only allows for semi-quantitative analysis. Moreover, only the mono-ubiquitinated species shows up as a distinct band, species with multiple ubiquitinations are detected in a smear.

CHEMICAL CROSS-LINKING COULD INCREASE THE LEVEL OF TBUBL1 IN THE IP IN FIG. 7C.

We agree that in principle cross-linking could increase the recovery of TbUbl1 in the pull down. However, we tried this quite extensively and were not able to increase the yield.

MINOR POINTS

- ATOM69 DEPLETION SHOWED RELATIVELY LOWER EFFECT ON DEPLETION OF MITOCHONDRIAL PROTEINS IN COMPARISON TO ATOM40 AND ATOM14 RNAI BUT INCREASED MORE NON-MITOCHONDRIAL PROTEINS IN THE CRUDE MITOCHONDRIAL FRACTION. FROM THIS OBSERVATION AUTHORS CONCLUDE THAT DEPLETION OF ATOM69 TRIGGERS A SPECIFIC STRESS RESPONSE (LINE 150-151). AT THIS STAGE OF WORK THIS STATEMENT IS AN OVER INTERPRETATION.

We agree and have rephrased the sentence.

- TB927.9.7200 HAS MTS, WHY IT IS EXCLUDED FROM THE MITO REFERENCE PROTEIN?

We have used the Importome as our reference for the mitochondrial proteome⁶. Tb927.9.7200 did not fulfill the criteria to be considered as a mitochondrial protein in this study. It was only quantified in two of four replicates, and had a p-value > 0.05. However, we have now tagged the protein at its C-terminus and show that it is indeed localized to mitochondria. See supplementary Figure 3 in the revised manuscript.

- LINE 129, A LINEAR MODEL

Corrected

REVIEWER #2 (REMARKS TO THE AUTHORS)

MITOCHONDRIA IMPORT MOST OF THEIR PROTEINS POST-TRANSLATIONALLY.

DEFECTS IN THE IMPORT MACHINERY CAN LEAD TO THE ACCUMULATION OF MITOCHONDRIAL PRECURSOR PROTEINS IN THE CYTOSOL. TO AVOID TOXIC EFFECTS OF THESE PRECURSORS, CELLS RECOGNIZE PRECURSORS AND RAPIDLY DEGRADE THEM. DESPITE ITS IMPORTANCE FOR CELLULAR FUNCTIONALITY, THE MOLECULAR BIOLOGY OF THIS DEGRADATION PROCESS IS NOT KNOWN. IN PARTICULAR, IT REMAINS UNCLEAR HOW CELLS CAN DISTINGUISH TOXIC NON-PRODUCTIVE PRECURSORS THAT HAVE TO BE DEGRADED FROM PRODUCTIVE PRECURSORS THAT ARE ON TRANSIT TO MITOCHONDRIA. THE STUDY BY DEWAR AND COLLEAGUES SHOWS A BEAUTIFUL MECHANISM WHICH RESULTS IN THE SPECIFIC DEPLETION OF CYTOSOLIC PRECURSORS FROM CELLS OF THE MODEL ORGANISM *TRYPANOSOMA BRUCEI*. THIS PARASITE EMPLOYS CO-FACTORS OF THE UBIQUITIN-PROTEASOME SYSTEM, IN PARTICULAR A PROTEIN WHICH THE AUTHORS CALLED TBUBL1, THAT RESIDES IN THE NUCLEUS UNDER NON-STRESS CONDITIONS. UPON ACCUMULATION OF MITOCHONDRIAL PRECURSORS, TBUBL1 IS RELEASED FROM THE NUCLEUS AND INDUCES THE UBIQUITINATION AND DEGRADATION OF PRECURSOR PROTEINS. BY USE OF VERY COMPREHENSIVE PROTEOME ANALYSES IN CELLS IN WHICH IMPORT COMPONENTS WERE DEPLETED, THE AUTHORS IDENTIFIED A LARGE SET OF PROTEINS WHICH ARE SPECIFICALLY RECRUITED ONTO THE MITOCHONDRIAL SURFACE. THIS STUDY IS VERY INSPIRING AND EXCITING. THE CLOSE COOPERATION OF THE CYTOSOL AND THE NUCLEUS IN THE MITOCHONDRIAL QUALITY CONTROL WAS RECENTLY REPORTED IN YEAST (SHAKYA, 2021. ELIFE 10:E61230). HOWEVER, THIS STUDY HERE GOES FAR BEYOND ANY PUBLISHED ARTICLES AS FOR THE FIRST TIME, IT REPORTS ABOUT A MECHANISM BY WHICH MITOCHONDRIAL PRECURSORS CAN BE SPECIFICALLY CLEANSED FROM STRESSED CELLS. TWO MAJOR ASPECTS SHOULD BE ADDRESSED IN ORDER TO EVEN IMPROVE THIS EXCITING STUDY:

1. ONE MAJOR OBSERVATION OF THIS STUDY IS THE RECRUITMENT OF PQC FACTORS TO THE MITOCHONDRIAL SURFACE UPON KNOCK DOWN OF ATOM SUBUNITS. HOW THESE FACTORS ARE DIRECTED TO THE OUTER MEMBRANE IS UNCLEAR. IT IS POSSIBLE THAT THIS RECRUITMENT IS MEDIATED SIMPLY BY THE BINDING OF THESE FACTORS TO THE TRANSLOCATION INTERMEDIATES THAT ARE STALLED ON THE OUTER MEMBRANE. ALTERNATIVELY, THEY DIRECTLY BIND THE OUTER MEMBRANE AS A RESULT OF A STRESS RESPONSE. IN ORDER TO DISTINGUISH THIS, THE AUTHORS MIGHT BLOCK PROTEIN SYNTHESIS PRIOR TO THE EXPERIMENT, E.G. FOR TWO HOURS BEFORE MITOCHONDRIA OF ATOM69-DEPLETED CELLS ARE ISOLATED. UNDER THESE CONDITIONS, TRANSLOCATION INTERMEDIATES SHOULD BE LARGELY ABSENT AND THUS, DIRECT BINDING AND PRECURSOR-MEDIATED RECRUITMENT COULD BE DISTINGUISHED. SUCH AN EXPERIMENT SHOULD BE ANALYZED BY MS RATHER THAN BY WESTERN BLOTTING IN ORDER TO GET A GENERAL OVERVIEW.

We did the suggested experiment (see Fig. 3 below). Using SILAC-MS, we compared crude mitochondrial fractions of induced ATOM69 RNAi cell lines that were incubated in the absence or presence of cycloheximide (CHX) for 2 hours (left panel) or 6 hours (right panel), respectively. The results show that the levels of most mitochondrial proteins in induced ATOM69 RNAi cells are slightly reduced in the presence of CHX, as would be expected since translation of new precursor proteins is abolished. Moreover, the recruitment of our candidate MQC factors is more pronounced in the presence of CHX, especially at the 6-hour time point. This would in principle suggest that putative MQC factors are directly recruited to the outer membrane and that their recruitment does therefore not directly depend on the presence of putative translocation intermediates. However, we also see a massive increase of non-mitochondrial proteins that are recruited to the crude mitochondrial fraction that depends on the time of the CHX treatment. The enriched proteins include many cytosolic ribosomal proteins as well as proteasomal subunits. We cannot explain this at the moment but believe the data is complicated by the fact that inhibition of translation causes many more alterations than just affecting the specific MQC we are studying. Thus, while we find these results very interesting we would prefer to not include them in the revised manuscript at the present time.

Figure 3. SILAC-MS analysis comparing induced ATOM69 RNAi cells in the absence and presence of cycloheximide (CHX). Candidate MQC factors are indicated in red. ATOM is indicated in green. Mitochondrial and non-mitochondrial proteins are depicted in black and grey, respectively.

2. THE EFFECT OF THE NUCLEAR EXPORT INHIBITOR LMB AS SHOWN IN FIG. 5 IS IMPRESSIVE. UNFORTUNATELY, THE AUTHORS ONLY USED THIS NICE DRUG FOR A WESTERN BLOT WITH FTSH-HA. THE STUDY WOULD BE EVEN MORE INTERESTING, IF THE AUTHORS WOULD PERFORM MS OF MITOCHONDRIAL FRACTIONS DERIVED FROM THESE LMB-TREATED CELLS. THEREBY THE SPECTRUM OF PROTEINS THAT ARE DEGRADED BY THE TBUBL1-DEPENDENT MECHANISM COULD BE IDENTIFIED. THIS WOULD BE A NICE, THOUGH NOT ESSENTIAL EXPERIMENTS. IT ALSO MIGHT HELP TO EXCLUDE THAT THE SUPPRESSION OF THE FTSH-HA DEGRADATION OBSERVED IS JUST THE CONSEQUENCE OF THE INEFFICIENT KNOCK-DOWN OF ATOM69 IN THE PRESENCE OF LMB.

We did the suggested SILAC-MS experiment and could identify 55 additional putative TbUbl1-dependent substrates. Moreover, as suggested by the reviewer we were able to exclude inefficient RNAi as a possible confounding factor.

The results of the new experiment are shown in new supplementary Fig. 10 and are discussed in a new paragraph (lines 335-342) in the revised manuscript.

MINOR POINTS:

3. THE NOMENCLATURE OF PROTEINS IN TB IS REALLY ANNOYING AS IT IS VERY DIFFICULT TO REMEMBER THESE NUMBERS. FOR PROTEINS STUDIED HERE, I RECOMMEND TO COME UP WITH AN ACRONYM THAT CAN BE USED BY OTHERS. SOMETHING LIKE MQC AND THE MOLECULAR MASS OF THE PROTEIN. ALSO THE ABBREVIATION OF UBL1 IS NOT PERFECT AS IT MIGHT BE CONFUSED WITH A UBIQUILIN WHICH HAS A RATHER SIMILAR FUNCTION IN HUMAN CELLS.

We considered the suggestion but decided to stay with the name. We don't want to name the proteins MQCxx since each protein may well have other as yet unknown functions as well.

4. THE DEPLETION OF MITOCHONDRIAL PROTEINS AS SHOWN IN THE DATA OF FIG. 1BC IS IMPRESSIVE AND VERY CONVINCING. SINCE IT IS VERY DIFFICULT TO HAVE A CLOSER LOOK AT THE NON-ATOM PROTEINS FOR SCIENTISTS WHO ARE NOT FAMILIAR WITH THE TRYPANOSOME PROTEIN NOMENCLATURE, IT WOULD BE INTERESTING TO ADD A FIGURE TO THE SUPPLEMENT IN WHICH THE AUTHORS ANALYZED, WHETHER THE DEPLETION PATTERN FROM THESE DIFFERENT SIRNAS ARE SIMILAR OR ALL DIFFERENT, FOR EXAMPLE BY A HEAT MAP OF SPEARMAN

CORRELATIONS. MOREOVER IT WOULD INTERESTING TO KNOW WHICH CLASS OF PROTEINS WERE PREDOMINANTLY DEPLETED? THE DATA SET WOULD ALLOW CONCLUSION ON WHETHER OUTER MEMBRANE PROTEINS ARE MORE DEPLETED THAN MATRIX PROTEINS, OR WHETHER MEMBRANE PROTEINS ARE MORE AFFECTED THAN HYDROPHILIC PROTEINS ETC. THIS WOULD BE AN INFORMATION OF GENERAL IMPORTANCE FOR A BROAD READERSHIP.

As suggested by the reviewer, we performed hierarchical clustering of mitochondrial proteins with altered abundance upon depletion of ATOM subunits. To reveal in particular potential differences in mitochondrial protein groups affected by the two receptors ATOM46 and ATOM69, we tested different parameters for clustering. Based on this, we applied a p-value threshold of 0.01 and removed SILAC data from RNAi experiments targeting ATOM40 and ATOM14 because their depletion resulted in decomposition of the ATOM complex and showed overall stronger effects (Fig. 1b). Hierarchical clustering of mitochondrial proteins affected in at least one of the remaining five ATOM subunits (ATOM19, ATOM12, ATOM11, ATOM46 and ATOM69) resulted in 5 clusters. Based on these data we can differentiate between mitochondrial protein groups that are mainly affected by one of the two receptor subunits or both receptors. Mitochondrial proteins affected by ATOM69-RNAi are found in cluster 2, 3 and 5. ATOM46-RNAi affected proteins are in cluster 4 and 5. Besides the notion that both receptors likely share many substrates, we recently could show that ATOM46 has a preference for binding more hydrophilic substrates by electrostatic interactions, whereas ATOM69 prefers to bind more hydrophobic substrates with at least one transmembrane domain by hydrophobic interactions¹. Since depletion of ATOM11 leads to a considerable reduction in the levels of both ATOM46 and ATOM69, it appears plausible that mitochondrial proteins in cluster 3, 4 and 5 are also affected in RNAi experiments targeting ATOM11. Interestingly, proteins of OXPHOS complexes and mitochondrial ribosomes are predominantly present in cluster 3 and cluster 5. Since these proteins are highly expressed, aggregation of their precursors under conditions of impaired mitochondrial protein import and depletion of cytosolic chaperones might be a crucial factor for triggering the quality control pathway observed in ATOM69-RNAi experiments.

The results of the hierarchical clustering are shown in the new Supplementary Fig. 1 and described in the main text (lines 159-162) of the revised manuscript. Furthermore, the clustering data are included in Supplementary Data 1b and information about the clustering approach is included in Materials & Methods (lines 668-677).

5. FIG. 7B SHOWS A PROMINENT MODIFICATION OF DMTS-COXIV-MYC FROM WHICH THE AUTHORS SUGGEST IT IS MONOUBIQUITINATION. IT ALSO COULD BY SUMO, OR? SEE N. THE PROMINENT MONOUBIQUITINATED DMTS-COXIV-MYC. COULDN'T THIS BE SUMO (SEE PAASCH ET AL., JBC 293, 599F)? SINCE IT IS UNCLEAR WHICH MODIFICATION IT REALLY IS, THE AUTHORS SHOULD BETTER NOT LABEL IT AS MONOUB, OR PROVE ITS UBIQUITINATION.

In the experiment shown in former Fig. 7b (Fig. 8b of the revised manuscript) the samples of the were affinity purified using ubiquitin-binding Ubiquapture beads. According to manufacturer these beads should not bind SUMO. Thus, as the band in question gets enriched by this procedure we would like to stay with monoUb

6. IN THE ABSTRACT, THE AUTHORS SHOULD BETTER CHANGE THE LAST SENTENCE ,SUCH PATHWAYS ARE AN OBLIGATE FEATURE OF ALL MITOCHONDRIA' FOR ,SUCH PATHWAYS ARE AN OBLIGATE FEATURE OF ALL EUKARYOTES' AS THE COMPONENTS DESCRIBED HERE ACT IN THE CYTOSOL/NUCLEUS.

The sentence was changed accordingly

7. LINE 158: 'THESE PROTEINS ARE DEGRADED BY THE PROTEASOME WHEN THEIR IMPORT IS INHIBITED'. IT IS ALSO POSSIBLE THAT LOWER AMOUNTS OF THESE PROTEINS WERE SYNTHESIZED. SUCH A REPRESSION OF MITOCHONDRIAL PROTEINS WAS ALSO OBSERVED IN OTHER SYSTEMS, SUCH AS IN YEAST (E.G. BOOS ET AL, NATURE CELL BIOLOGY).

We have toned down the phrase in question.

REVIEWER #3 (REMARKS TO THE AUTHOR):

SUMMARY: IN THIS REPORT BY DEWAR ET AL., THE AUTHORS INVESTIGATE THE EFFECTS OF KNOCKING DOWN THE ATOM69 SUBUNIT OF THE MITOCHONDRIAL PROTEIN IMPORT RECEPTOR COMPLEX IN T BRUCEI. DEPLETION OF ATOM69 UNCOVERED A PROTEIN QUALITY CONTROL PATHWAY THAT LIKELY FUNCTIONS IN NORMAL CELLS AND LED TO THE IDENTIFICATION OF THREE PUTATIVE COMPONENTS OF THIS PATHWAY. WHILE MITOCHONDRIAL PROTEIN QUALITY CONTROL MECHANISMS HAVE BEEN UNCOVERED IN OTHER MODEL ORGANISMS INCLUDING YEAST, NEMATODES AND MAMMALS, FINDING SIMILAR PATHWAYS IN THE HIGHLY DIVERGED TRYPANOSOME SUGGEST THAT MITOCHONDRIAL QUALITY CONTROL IS A CENTRAL FEATURE OF ALL EUKARYOTES. THIS WORK IS SIGNIFICANT AND OF INTEREST TO A WIDE RANGE OF CELL BIOLOGISTS, WARRANTING PUBLICATION IN NATURE COMMUNICATIONS.

SPECIFIC COMMENTS:

1. THE AUTHORS CONCLUDE THAT ABLATION OF ATOM69 TRIGGERS A SPECIFIC STRESS RESPONSE NOT SEEN WHEN THE OTHER SIX SUBUNITS OF THE ATOM COMPLEX (ATYPICAL TRANSLOCASE OF THE OUTER MEMBRANE) ARE KNOCKED DOWN. THIS RESULT IS PUZZLING SINCE LOSS OF MANY OF THE OTHER ATOM SUBUNITS ALSO SHOW REDUCED OR ABSENT ATOM69 LEVELS, ONE MIGHT EXPECT THAT A SIMILAR STRESS PATHWAY IS LIKEWISE ACTIVATED. WHILE THEIR PROTEOMIC ANALYSES CLEARLY SHOWS THAT LOSS OF THE OTHER ATOM SUBUNITS DOES NOT LEAD TO MITOCHONDRIAL ACCUMULATION OF TBUBL1, TBE3HECT1, OR TB927.9.7200, IT DOES NOT RULE OUT THAT THESE PROTEINS ARE ALSO REQUIRED FOR THE DEGRADATION OF NON-IMPORTED MITOCHONDRIAL PROTEINS WHEN THE OTHER ATOM SUBUNITS ARE DEPLETED. THE AUTHORS DO ADDRESS THIS, ALBEIT VERY BRIEFLY, IN THE DISCUSSION.

We agree and have modified and extended the paragraph in question in the discussion (see lines 504-514).

PERHAPS DETERMINING DIRECTLY WHETHER TBFTSH-MYC TURNOVER REQUIRES TBUBL1, TBE3HECT1, OR TB927.9.7200 IN ATOM11, 12, 40 OR 46 KNOCKDOWNS IS WARRANTED.

The suggested experiments would involve the production of at least 12 novel cell lines all of which would require multiple rounds of transfections. We believe this is beyond the scope of the present work. - However, we think that simpler experiments can address the issue regarding the specificity of the studied MQC pathway. A hallmark of the described pathway is the nuclear release of TbUbl1. Thus, we tested whether this release also occurs in induced ATOM12 and ATOM46 RNAi cell lines in which the level of ATOM69 is not affected. The results in the new Figure 4bc of the revised manuscript strongly suggests that this is not the case. (See new paragraph in the revised manuscript: lines 277-288).

Furthermore, we performed a complementary experiment. Instead of removing the entire ATOM69 receptor, we replaced wildtype ATOM69 with a version of the receptor that lacks the N-

terminal CS/Hsp20-like chaperone-binding domain. The new Figure 4bc of the revised manuscript shows that in the resulting cell line a significant release (51%) of TbUbl1 to the cytosol is observed. This suggests that the lack of the chaperone-binding domain alone is sufficient to trigger the release of TbUbl1 from the nucleus. Intriguingly, the cell line did not show a strong growth phenotype, indicating that the MQC pathway can digest the small amount of cytosolic protein aggregates that may have formed in the absence of the chaperone binding domain. In line with this the levels of FtsH were not reduced in the cell line. (See new paragraph in the revised manuscript: lines 289-304).

2. ALONG SIMILAR LINES, THE AUTHORS PROPOSE THAT ATOM69, LIKE THE YEAST TOM70 COUNTERPART, IS REQUIRED FOR THE IMPORT OF A SUBSET OF MITOCHONDRIAL PROTEINS—SUCH AS HYDROPHOBIC OR AGGREGATION-PRONE SUBSTRATES. HOWEVER, TO PROBE THE FUNCTION OF ATOM69 AND TBUBL1 THEY USE THE SMALL, HYDROPHILIC SUBSTRATE, COXIV, WHICH BASED ON THEIR MODEL IS NOT IMPORTED VIA THE ATOM69 PATHWAY. WHILE THEIR RESULTS INDEED SHOW THAT LOSS OF TBUBL1 LEADS TO STABILIZATION (AND UBIQUITINATION) OF COXIV, IT SEEMS TO ARGUE AGAINST THEIR IDEA OF AN ATOM69-SPECIFIC PATHWAY.

We do not claim that ATOM69 or ATOM46 have mutually exclusive substrate specificities, rather we always talk about substrate preferences. The reason is that in vivo import of most proteins depends to various extents on both receptors ¹. Thus, there is indeed a large overlap. A recent study in yeast suggests that the situation is similar for yeast Tom70 which in vitro has a preference for hydrophobic substrates but in vivo is also required to import some presequence-containing proteins ². Previous work from our lab has shown that import of COXIV depends to essentially equal parts on ATOM69 (50.3%) and ATOM46 (49.7%) ¹.

The reviewer is correct that COXIV is a soluble protein lacking transmembrane domains. However, when we analyzed its properties experimentally we found it does not behave like a typical soluble protein. In a carbonate extraction at high pH the majority of COXIV fractionates with the pellet, even though soluble and peripheral membrane proteins are expected to be completely soluble under these conditions. Moreover, in a digitonin-based aggregation assay as described in ³, more than 30% of both native as tagged COXIV aggregate in 1% digitonin, whereas much less aggregation (below 13%) was seen for two integral membrane proteins (VDAC and TimRhom I) or two soluble proteins (Hsp60 and Alba1). Thus, while COXIV lacks transmembrane domains it is more aggregation prone than other soluble and even integral membrane proteins tested. This may explain why it requires both receptors for import.

We have added the results of experiment described above as new supplementary Fig. 11 to the revised manuscript (described in lines 343-354). Moreover, we have modified Fig. 6bc and Fig. 7bc and added the data for COXIV so that it can be compared to our other substrate FtsH.

3. TO SHOW THAT NUCLEAR RELEASE OF TBUBL1 IS REQUIRED FOR MITOCHONDRIAL PROTEIN TURNOVER, THE AUTHORS ADD A SECOND NLS TO TBUBL1 AND OVEREXPRESS THIS CONSTRUCT IN CELLS. CONSISTENT WITH THEIR EXPERIMENTS INHIBITING EXPORT (WITH LMB), THE TBUBL1-NLS CONSTRUCT INHIBITS TURNOVER. THE AUTHORS CONCLUDE THAT THEY HAVE CONSTRUCTED A DOMINANT-NEGATIVE VERSION OF TBUBL1, BUT THIS IS NOT THE ONLY EXPLANATION FOR WHAT IS GOING ON. MY LEAST FAVORITE PART OF THE MANUSCRIPT OVERALL IS THAT THE AUTHORS SEEM BOUND AND DETERMINED TO FIT THEIR RESULTS INTO THE MITOCHONDRIAL PROTEIN QUALITY CONTROL 'BOX', INSTEAD OF LETTING THE RESULTS LEAD WHERE THEY MAY. NONETHELESS, I AM CONFIDENT THAT THEY ARE MOST LIKELY CORRECT IN THEIR INTERPRETATIONS AND THAT THIS WORK, WITH A BIT OF MOSTLY EDITING WORK, IS SUITABLE FOR PUBLICATION.

We would like to emphasize that our interpretation of the experiment shown in Fig. 7 in the revised manuscript) is influenced by the independent experiment depicted Fig. 6 in the revised manuscript) where nuclear export was inhibited by LMB. The simplest interpretation of both experiments is, in our opinion, that the release of TbUbl1 from the nucleus is a prerequisite for the proteasomal digestion of our model substrates FtsH and COXIV.

4. PICKY POINT: THE AUTHORS USE THE TERM DOWN REGULATED IN SEVERAL PLACES IN THE MANUSCRIPT. SINCE THIS TERM IS LIKELY TO BE MISINTERPRETED BY MANY TO INDICATION TRANSCRIPTIONAL REGULATION, IT IS PROBABLY BEST TO AVOID.

We would like to stay with the word regulated. If we replace it by terms like “reduced in level” the text becomes less readable. We think the term regulated will not be misinterpreted as transcriptional regulation since it is already mentioned in the title of the manuscript that MQC pathway studied is posttranscriptional.

References

- 1 Rout, S. *et al.* Determinism and contingencies shaped the evolution of mitochondrial protein import. *Proc Natl Acad Sci U S A* **118**, doi:10.1073/pnas.2017774118 (2021).
- 2 Backes, S. *et al.* The chaperone-binding activity of the mitochondrial surface receptor Tom70 protects the cytosol against mitoprotein-induced stress. *Cell Rep* **35**, 108936, doi:10.1016/j.celrep.2021.108936 (2021).
- 3 Poveda-Huertes, D. *et al.* An Early mtUPR: Redistribution of the Nuclear Transcription Factor Rox1 to Mitochondria Protects against Intramitochondrial Proteotoxic Aggregates. *Mol Cell* **77**, 180-188 e189, doi:10.1016/j.molcel.2019.09.026 (2020).
- 4 Kovalinka, T., Panek, T., Kovacova, B. & Horvath, A. Orientation of FtsH protease homologs in *Trypanosoma brucei* inner mitochondrial membrane and its evolutionary implications. *Mol Biochem Parasitol* **238**, 111282, doi:10.1016/j.molbiopara.2020.111282 (2020).
- 5 Mani, J. *et al.* Mitochondrial protein import receptors in Kinetoplastids reveal convergent evolution over large phylogenetic distances. *Nat Commun* **6**, 6646, doi:10.1038/ncomms7646 (2015).
- 6 Peikert, C. D. *et al.* Charting Organellar Importomes by Quantitative Mass Spectrometry. *Nat. Commun.* **8**, 15272 (2017).

Peer review comments, second round review –

Reviewer #1 (Remarks to the Author):

The authors responded adequately for all the concerns raised by the reviewers

Reviewer #2 (Remarks to the Author):

The authors satisfactorily addressed all points raised on the previous version. I would recommend to include the control experiment with the cycloheximide treatment and show it in the supplements but leave the decision to the authors and the editor.

I fully support publication of this exciting study in its present form.

Reviewer #3 (Remarks to the Author):

I am satisfied by the authors' revisions and recommend publication at this time